# Proteomics of CKD progression in the chronic renal insufficiency cohort

Ruth F. Dubin [1,19] ✉, Rajat Deo[2,19], Yue Ren[3], Jianqiao Wang[4], Zihe Zheng [3], Haochang Shou [3], Alan S. Go[5], Afshin Parsa[6], James P. Lash[7], Mahboob Rahman[8], Chi-yuan Hsu[5,9], Matthew R. Weir[10], Jing Chen[11], Amanda Anderson[11], Morgan E. Grams [12,13,14], Aditya Surapaneni[12,13,14], Josef Coresh [12,13], Hongzhe Li[3], Paul L. Kimmel[15], Ramachandran S. Vasan[16], Harold Feldman[3], Mark R. Segal[17], Peter Ganz [18], CRIC Study Investigators* & CKD Biomarkers Consortium*

Progression of chronic kidney disease (CKD) portends myriad complications, including kidney failure. In this study, we analyze associations of 4638 plasma proteins among 3235 participants of the Chronic Renal Insufficiency Cohort Study with the primary outcome of 50% decline in estimated glomerular filtration rate or kidney failure over 10 years. We validate key findings in the Atherosclerosis Risk in the Communities study. We identify 100 circulating proteins that are associated with the primary outcome after multivariable adjustment, using a Bonferroni statistical threshold of significance. Individual protein associations and biological pathway analyses highlight the roles of bone morphogenetic proteins, ephrin signaling, and prothrombin activation. A 65-protein risk model for the primary outcome has excellent discrimination (C-statistic[95%CI] 0.862 [0.835, 0.889]), and 14/65 proteins are druggable targets. Potentially causal associations for five proteins, to our knowledge not previously reported, are supported by Mendelian randomization: EGFL9, LRP-11, MXRA7, IL-1 sRII and ILT-2. Modifiable protein risk markers can guide therapeutic drug development aimed at slowing CKD progression.

Chronic kidney disease (CKD) affects 15% of the U.S. population[1]. Progression of CKD is associated with a high risk of medical complications including cardiovascular disease[2], bone and metabolic disease[3,4], and frailty[4]. Patients who progress to kidney failure need to consider initiation of dialysis or a kidney transplant. The cost of care for patients with advanced CKD adds a significant burden to the healthcare system[5]. Anticipating how rapidly a person with CKD will progress to kidney failure and discovering biomarkers of CKD progression and potential therapeutic targets for slowing CKD progression remain high priorities[6].

Proteins regulate biological processes and integrate the effects of genes with those of the environment, age, comorbidities, behaviors,

and drugs[7–9]. Multiprotein models predict the risk of developing diseases and their clinical outcomes as well or better than traditional clinical models[7–9]. The 4-variable Kidney Failure Risk Equation (KFRE)[4,10], the most commonly used tool for predicting CKD progression to kidney failure, consists of estimated glomerular filtration rate (eGFR), age, sex, and albuminuria. Whereas KFRE is highly predictive of progression to kidney failure with a c-statistic of ~0.88 at 5 years[10], among its components only albuminuria is readily modifiable with treatment. Personalized prognostic equations for CKD progression that consist of modifiable biological factors could be used to monitor responses to medical treatments. For example, a prognostic equation for cardiovascular risk that consisted of modifiable protein risk factors

A full list of affiliations appears at the end of the paper. *Lists of authors and their affiliations appear at the end of the paper.
✉e-mail: ruth.dubin@utsouthwestern.edu

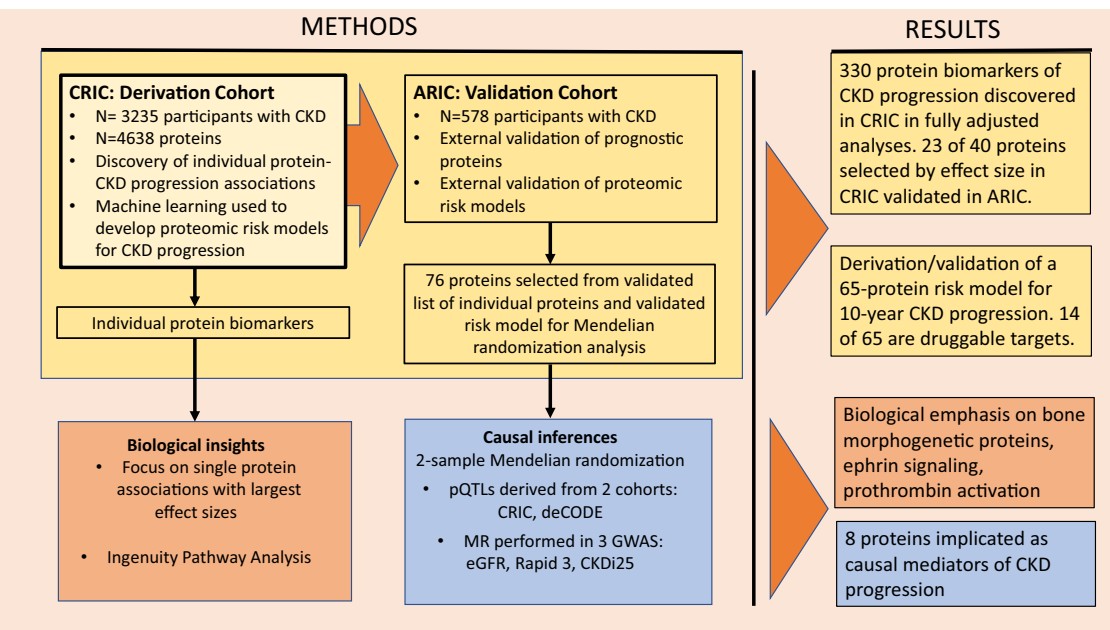

**Fig. 1 | Summary of study design and results.** The study design, including derivation and validation for risk models and individual proteins, and selection of proteins for pathway and Mendelian randomization, are illustrated above. Our results include novel risk models for CKD progression as well as biological insights into potential causal mediators of kidney disease.

accurately predicted which patients remained at high risk for poor outcomes and might benefit from more specialized therapies[11].

Niewczas and colleagues examined 194 circulating inflammatory proteins in a total of 525 participants with type 1 and type 2 diabetes and identified a kidney risk inflammatory signature (KRIS), consisting of 17 proteins enriched for the tumor necrosis factor receptor superfamily members, that was associated with a 10-year risk of kidney failure[12]. More recently, the same group of investigators measured 1129 plasma proteins in a total of 358 participants with type 1 and type 2 diabetes and identified 3 proteins associated with a lower risk that are potentially protective against the progression of CKD to kidney failure[13].

In this study, we have utilized SomaScan V.4.0 (SomaLogic, Boulder, CO), a large-scale aptamer proteomic platform that measures nearly 5000 distinct plasma proteins simultaneously, to conduct the largest proteomic analysis of CKD progression to date. Our derivation cohort consisted of 3235 participants of the Chronic Renal Insufficiency Cohort (CRIC). By design, CRIC includes nearly equal numbers of participants with and without diabetes[14]. The validation cohort consisted of 578 participants with CKD (eGFR<60 ml/min/1.73 m²) from the Atherosclerosis Risk in Communities Study (ARIC). Our goals were (1) to discover numerous plasma proteins that are markers or mediators of CKD progression; (2) to identify biological pathways leading to CKD progression; (3) to elucidate whether protein markers of CKD progression differ by diabetic status or other clinical factors; and (4) to build a multiprotein prognostic model for CKD progression that is highly predictive and includes factors potentially more modifiable than those in the KFRE. A summary of the study design is illustrated in Fig. 1.

## Results

### CRIC cohort and renal outcomes
Detailed baseline characteristics are found in Supplementary Data 1. In brief, among the 3235 CRIC participants included in the analysis of the primary outcome, 10-year kidney failure/50% eGFR decline, mean (±SD) age was 59 (±11) years, eGFR was 43 (±17) ml/min/1.73 m², 45% were women, and by design, nearly 50% had a history of diabetes. There were a total of 1139 (35%) events, including 998 (31%) kidney

failure events, over median (IQR) 6.0 (2.6–10.0) years. Participants who reached the primary outcome were older, more likely to be male, black, diabetic, and have a lower baseline eGFR, higher albuminuria, and history of CVD. For the secondary outcome of 4-year eGFR slope, the median (IQR) eGFR slope was −1.01 (−2.18, 0.27) ml/min/1.73 m² per year; 316 (9.74%) had eGFR slope ≤ −3 (ml/min/1.73 m²)/year.

### ARIC validation cohort for the primary renal outcome
The validation cohort was comprised of 578 ARIC participants with eGFR <60 ml/min/1.73 m² at ARIC Visit 3, all of whose samples were assayed with the same version of SomaScan. These ARIC participants had a mean age of 64 years, a lower prevalence of diabetes (32%), and higher mean eGFR (48 ml/min/1.73 m²). There were 85 (15%) events for the primary renal outcome, kidney failure or a 50% decline in eGFR, including 80 kidney failure events (Supplementary Data 2).

### Associations of individual proteins with the primary outcome in CRIC and ARIC
Associations of individual proteins with the primary outcome (≥50% eGFR decline or kidney failure within 10 years) are visualized in Fig. 2 as Volcano plots, shown unadjusted (Fig. 2A), adjusted for eGFR (Fig. 2B) and fully adjusted (Fig. 2C). Among the 4638 proteins investigated, in fully adjusted analyses, 330 proteins (7.1% of all proteins measured) were associated with primary renal outcome at FDR significance ($q < 0.05$). We identified numerous proteins associated with a higher risk of the primary outcome. Whereas only 1 of the previously reported 17 KRIS proteins had fully adjusted $\log_2$ HR > 2, 14 additional proteins with fully adjusted HRs between 2 and 5 were identified in this study (Fig. 2C). The top 20 proteins with the largest HR per $\log_2$, listed with their biological functions and current drugs that target them, are shown in Table 1. We identified numerous proteins associated with lower risk (HR < 1), referred to in the literature as potentially protective, which are shown in Fig. 2 and Supplementary Data 6. Protein associations are shown as HR per MAD unit in the Supplementary Data (Supplementary Data 3–6, 9–12).

Higher-risk markers in CRIC and ARIC were enriched with members of the ephrin family (5 of 20 top proteins) and bone morphogenetic proteins (BMPs) (4 of 20 top proteins). Through a search of the

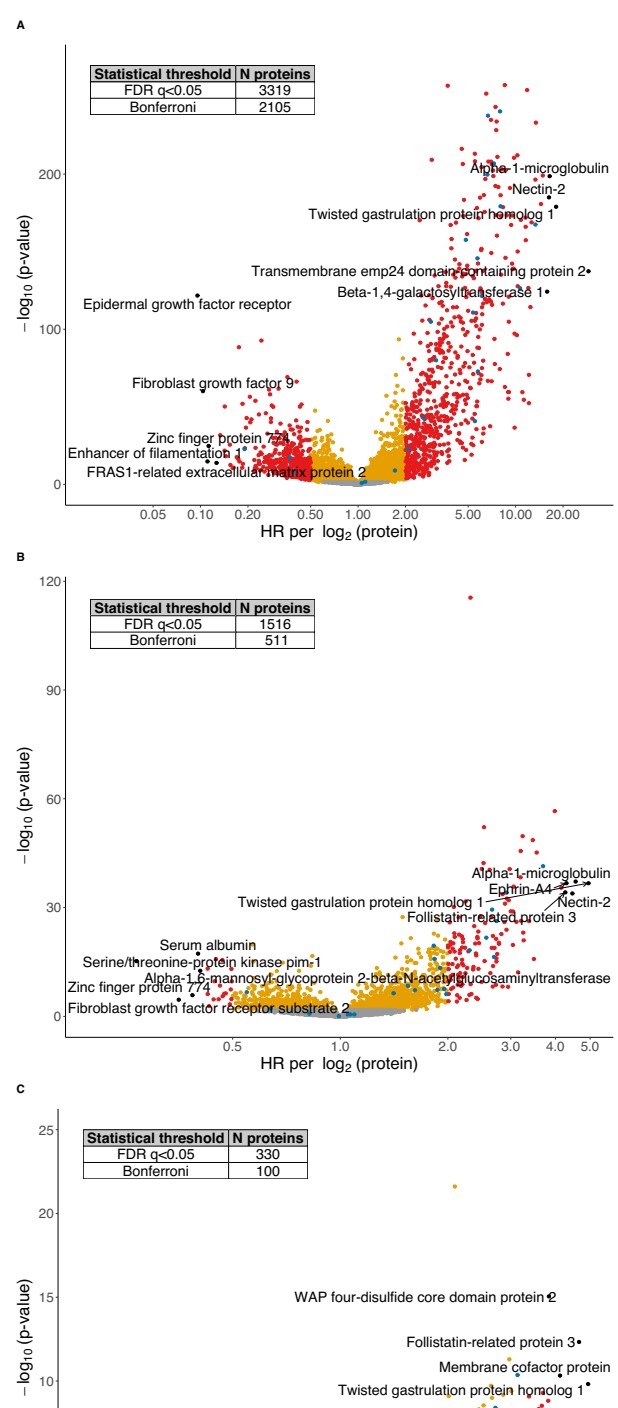

**Fig. 2 | Volcano plots of individual protein associations with the primary outcome.** Associations of 4638 proteins with the primary outcome unadjusted (**A**), eGFR adjusted (**B**) and after full adjustment for age, gender, race, eGFR, log[urine protein to creatinine ratio], systolic blood pressure, diabetes, smoking status, body mass index, and cardiovascular disease history (**C**). 17 proteins in the Kidney Risk Inflammatory Signature, and 3 proteins previously found to be associated with lower risk in patients with DM are labeled as blue dots. *P* values are two-sided. FDR false discovery rate<0.05. FRAS1 Fraser extracellular matrix complex subunit 1, WAP whey acid protein.

## Pathway analysis of proteins associated with CKD progression in CRIC

We performed an overrepresentation analysis with the IPA tool to define the canonical pathways linked to the primary outcome. There were 1516 proteins that were associated with the primary outcome of CKD progression at an FDR of 5%, after adjustment for eGFR, and we compared this subset of proteins to the 4638 background proteins measured by SomaScan. The top ten canonical pathways are listed in Table 2. Ephrin signaling was again prominent, represented as the Ephrin A pathway. There was also significant enrichment for proteins that link inflammation and metabolic processes (LXR/RXR activation), matrix metalloprotease inhibition, hepatic fibrosis, and intrinsic prothrombin activation pathway. In "Discussion", we focus on potential roles for ephrin signaling, BMPs and prothrombin activation pathways in worsening kidney disease.

## Mendelian randomization

Using CRIC genotype data, we identified one or more pQTLs for 23 of 76 of our selected protein risk factors. Within the eGFR database, we found significant MR associations for four proteins (listed with SNP and *P* value): protein delta homolog 2/EGFL9 (rs2125739, $1.9 \times 10^{-6}$), low-density lipoprotein receptor-related protein 11/LRP-11 (rs9689036, $1.56 \times 10^{-4}$), Interleukin-1 receptor type 2/IL-1 sRII (rs2310170, $1.3 \times 10^{-3}$), and alpha-1 microglobulin (rs10982054, $1.7 \times 10^{-3}$). Within the CKDi25 GWAS, one variant was significant: EGF-containing fibulin-like extracellular matrix protein 1 (aka fibulin 3 FBLN3) (rs6755214, $1.3 \times 10^{-3}$). None of the variants we tested had MR associations in the Rapid3 database. Using the deCODE database, 54 of 76 proteins were linked to cis pQTLs. Significant MR associations were confirmed for LRP-11 in the eGFR GWAS, and for FBLN3 in the CKDi25 GWAS. In addition, significant MR associations were found for matrix metalloproteinase 7/MMP-7 (Rapid3 GWAS: 7 SNPs comprised instrumental variable (IV), $P = 7.0 \times 10^{-4}$), leukocyte immunoglobulin-like receptor subfamily B member 1/ILT-2 (Rapid3: 5 SNPs comprised IV, $P = 1.6 \times 10^{-6}$) and matrix remodeling associated 7/MXRA7 (Rapid3: 2 SNPs comprised IV, $P = 1.2 \times 10^{-5}$) (Fig. 3). MR associations with nominal $P < 0.05$ were observed for several proteins including two BMP antagonists, follistatin-related protein 3 (FSTL3) and twisted gastrulation protein homolog 1 (TWSG1), as well as CILP2, a protein associated with a lower risk of CKD progression that inhibits fibrosis, all of which replicated in ARIC. Two proteins that passed MR with significance after adjustment for multiple testing are druggable targets: IL-1 sRII and MMP-7 (Druggable Target Database)[15] (Supplementary Data 7).

## Impact of diabetic status for 20 previously reported risk factors

We visualized HRs of all proteins we found to be significantly associated with the primary outcome in a scatterplot, comparing participants with vs. without DM. HRs were similar in direction and effect size for DM vs. non-DM for the primary outcome (*rho* = 0.68) and the kidney failure outcome alone (*rho* = 0.68) (Supplementary Information Fig. 1). In addition, we examined 17 KRIS proteins reported by Niewczas et al. to predict the progression of CKD to kidney failure in cohorts of patients with diabetes as well as three proteins associated with lower

Druggable Target Database[15], 8 of these 20 proteins are currently druggable targets (Table 1). The three proteins with the lowest HRs that passed the criteria for validation in ARIC are Cartilage intermediate layer protein 2 (CILP2), C1GALT1-specific chaperone 1, and albumin. Potential roles for these proteins in the biology of CKD progression are delineated in "Discussion".

**Table 1 | Individual proteins associated with the CKD progression primary outcome in both CRIC and ARIC studies**

| Protein | Adjusted for eGFR HR in CRIC | | Fully adjusted HR in CRIC | | Biological function | Relevant drug |
|---|---|---|---|---|---|---|
| | HR (95% CI) | P | HR (95% CI) | P | | |
| *Twisted gastrulation protein homolog 1* | 4.95 (3.87, 6.32) | 1.89e-37 | 2.8 (2.04, 3.84) | 1.52e-10 | Antagonist of bone morphogenetic proteins— such as BMP 7, which is protective against renal fibrosis[63] | |
| Alpha-1-microglobulin | 4.55 (3.61, 5.74) | 6.59e-38 | 1.41 (1.09, 1.83) | 0.0083 | Antioxidant, binds and degrades heme | |
| Ephrin-A4 | 4.29 (3.43, 5.36) | 1.85e-37 | 2.29 (1.76, 2.98) | 5.18e-10 | Protein-tyrosine kinase | CAR T-cell therapy |
| *Follistatin-related protein 3* | 4.26 (3.38, 5.36) | 6.31e-35 | 2.69 (2.06, 3.52) | 4.68e-13 | Antagonist of bone morphogenetic proteins[64] such as GDF11 | |
| Low-density lipoprotein receptor-related protein 11 | 4.15 (3.32, 5.18) | 3.34e-36 | 2.05 (1.6, 2.61) | 1.01e-08 | Membrane protein related to lipid metabolism | |
| Ephrin type-A receptor 2 | 3.98 (3.36, 4.72) | 2.48e-57 | 1.98 (1.63, 2.40) | 5.07e-12 | Protein-tyrosine kinase; ongoing research to develop ephrins as therapeutic target[65] | CAR T-cell therapy |
| Tumor necrosis factor receptor superfamily member 1A | 3.69 (3.06, 4.46) | 3.85e-42 | 2.05 (1.66, 2.54) | 4.40e-11 | Inflammatory mediator; predicts ESRD[12] and incident CKD[66] | VB-111 |
| WAP four-disulfide core domain protein 2 | 3.54 (2.98, 4.22) | 6.50e-46 | 2.35 (1.91, 2.90) | 8.87e-16 | Glycoprotein; protease inhibitor. Also called Human epididymis protein 4 or MMP2. Mediator of renal fibrosis[67–69] | |
| Matrix remodeling-associated protein 7 | 3.45 (2.93, 4.07) | 2.46e-49 | 1.82 (1.51, 2.19) | 1.93e-10 | ECM protein involved in kidney fibrosis[70] | |
| Erythropoietin receptor | 3.24 (2.77, 3.77) | 2.03e-50 | 1.73 (1.44, 2.07) | 4.85e-09 | Cytokine receptor. Active in a variety of tissues, including bone formation[71] | |
| Tyrosine-protein kinase transmembrane receptor ROR2 | 3.2 (2.73, 3.75) | 2.44e-46 | 1.83 (1.51, 2.22) | 1.01e-09 | Protein-tyrosine kinase | CCT301-38 (targeting ROR2) |
| Ephrin-A2 | 3.11 (2.53, 3.83) | 8.41e-27 | 1.93 (1.53, 2.43) | 3.38e-08 | Protein-tyrosine kinase | CAR T-cell therapy |
| *Brorin* | 3.04 (2.51, 3.69) | 1.19e-29 | 1.61 (1.31, 1.98) | 4.95e-06 | Antagonist of bone morphogenetic proteins; antagonists of BMP modulate the ECM[18] | |
| Netrin receptor UNC5B | 3.0 (2.48, 3.63) | 1.07e-29 | 1.75 (1.40, 2.17) | 5.54e-07 | Inhibits vascular branching during angiogenesis; active in diabetic kidney[72] | |
| Vesicular integral-membrane protein 36 | 2.92 (2.38, 3.58) | 1.27e-24 | 1.53 (1.20, 1.95) | 5.27e-04 | Transport and secretion of glycoproteins | |
| Ephrin type-B receptor 4 | 2.89 (2.42, 3.45) | 8.054e-32 | 1.69 (1.38, 2.08) | 4.27e-07 | Protein-tyrosine kinase; stimulates angiogenesis after kidney injury to enhance recovery[27] | KD019 |
| Ephrin-B2 | 2.65 (2.23, 3.15) | 4.41e-28 | 1.92 (1.56, 2.37) | 6.69e-10 | Protein-tyrosine kinase; stimulates angiogenesis after kidney injury to enhance recovery[27] | BVD-523 |
| Protein delta homolog 2 | 2.62 (2.27, 3.01) | 3.66e-41 | 1.68 (1.41, 2.01) | 1.27e-08 | Soluble form inhibits adipogenesis; may antagonizes NOTCH pathway[32] | |
| *Neuroblastoma suppressor of tumorigenicity 1* | 2.52 (2.24, 2.84) | 6.68e-53 | 1.46 (1.27, 1.68) | 5.89e-08 | Antagonist of bone morphogenetic proteins, associated with progression to ESRD[73] | |
| Hepatitis A virus cellular receptor 1/KIM-1 | 2.31 (2.15, 2.48) | 3.332e-116 | 1.55 (1.42, 1.7) | 2.45e-22 | Marker of acute kidney injury[74] | |

CAR T Chimeric antigen receptor T cell, KIM-1 kidney injury molecule 1, ESRD end-stage renal disease, ECM extracellular matrix, LVH left ventricular hypertrophy, GDF growth differentiation factor, BMP bone morphogenetic protein, LDL low-density lipoprotein, MMP matrix metalloproteinase, WAP whey acid protein, ROR2 receptor-related 2, UNC5B uncoordinated 5B, CILP1 Cartilage intermediate layer protein 1, C1GALT1 Core 1 synthase, glycoprotein-N-acetylgalactosamine 3-beta-galactosyltransferase, 1, TGFB transforming growth factor beta.

HRs are expressed per log$_2$ of protein level and raw P values are two-sided. The fully adjusted model includes age, gender, race, eGFR, log [urine protein-to-creatinine ratio], systolic blood pressure, diabetes, smoking status, body mass index, and cardiovascular disease history. A druggable target search was performed using the Therapeutic Target Database at http://db.idrblab.net/. Bold italic font indicates proteins related to ephrin signaling; bold font indicates proteins related to bone morphogenetic proteins.

risk[12,13]. After full adjustment, six of these 20 proteins replicated in CRIC at $P < 0.0025$ (0.05/20) for the outcome of kidney failure among participants with DM. Three of these five KRIS proteins that replicated were members of the tumor necrosis factor receptor superfamily—members 1A, 1B (measured with two aptamers) and 19. One of the three lower-risk proteins, angiopoietin-1, replicated in the CRIC participants with DM. There was a significant interaction of DM with Interleukin-18 receptor 1 and angiopoietin-1: HR per $\log_2$ [95% CI] for Interleukin-18 receptor 1 in DM 1.35 [1.08, 1.69] vs non-DM 0.77 [0.57, 1.1], $P$ for interaction 0.008; HR [95% CI] for angiopoietin-1 in DM 0.78 [0.69, 0.87] vs non-DM 1.07 [0.90, 1.3], $P$ for interaction <0.001 (Supplementary Data 8).

**Table 2 | Canonical pathways among 1516 proteins associated with CKD progression**

| IPA canonical pathway | P value | Ratio |
|---|---|---|
| Intrinsic prothrombin activation pathway | 0.000019 | 0.68 (23/34) |
| Neuroprotective role of THOP1 in Alzheimer's disease | 0.0093 | 0.46 (31/67) |
| Hepatic fibrosis/hepatic stellate cell activation | 0.0095 | 0.43 (49/115) |
| LXR/RXR activation | 0.012 | 0.44 (37/84) |
| Axonal guidance signaling | 0.016 | 0.39 (91/236) |
| SPINK1 pancreatic cancer pathway | 0.018 | 0.49 (20/41) |
| Regulation of the epithelial mesenchymal transition in the development pathway | 0.023 | 0.55 (12/22) |
| Ephrin A signaling | 0.024 | 0.52 (14/27) |
| Extrinsic prothrombin activation pathway | 0.027 | 0.62 (8/13) |
| Inhibition of matrix metalloproteases | 0.029 | 0.52 (13/25) |

*IPA* Ingenuity Pathway Analysis, *THOP1* Thimet Oligopeptidase 1, *LXR/RXR* Liver X Receptor-Retinoid X Receptor, *SPINK1* serine peptidase inhibitor Kazal type 1.
For each pathway, the ratio indicates the number of proteins significantly associated with the primary outcome at FDR < 0.05 after adjustment for eGFR, divided by the number of proteins in that pathway measured in our study. Raw, one-sided *P* values are calculated using Fisher's exact test.

## Association of individual proteins with short-term kidney function decline

Twenty individual proteins associated with the highest and 20 with the lowest risk of 4-year eGFR decline are listed in Supplementary Data 9 and 10, respectively. Among those predicting a faster decline, 15/20 were also among the top 20 protein risk factors for the primary outcome, and 15/20 had associations with eGFR decline that remained significant after full adjustment at FDR < 0.05. These risk factors for eGFR decline included ephrin receptors and tumor necrosis factor receptor 1A. Less well-known risk factors included brorin and erythropoietin receptors, both of which were successfully validated in ARIC for the primary outcome. Among proteins predicting a slower decline, 15/20 were listed among protein factors with the lowest hazard ratio for the primary outcome. Three of these proteins remained significant by FDR *q* value after full adjustment: mitochondrial superoxide dismutase, fibroblast growth factor 9, and follistatin-related protein 5. For further description of specific proteins, see "Discussion".

## Risk prediction models for primary and secondary outcomes in CRIC

In the 80% training set of CRIC participants, we derived a 65-protein model for the primary outcome (≥ 50% eGFR decline or kidney failure within 10 years) using elastic net regression. The β-coefficients, adjusted HRs, and relevant drug for each protein included in the risk model are listed in Supplementary Data 11. In the 20% CRIC testing set, the model yielded a C-statistic of 0.862 (95% CI: 0.835, 0.888), with similar discrimination to the refit KFRE. Similarly, we derived a 20-protein model for the secondary outcome (4-year eGFR slope) in the 80% training set. The β-coefficients, adjusted HRs, and available drug for each protein included in the risk model are listed in Supplementary Data 12. In the 20% testing set, the C-statistic (95% CI) for the 20-protein model was 0.728 (0.708, 0.748), similar to the refit KFRE (0.744 (0.725, 0.763)). Hybrid clinical-protein models for both primary and secondary outcomes showed incremental, statistically significant improvement in discrimination over the refit KFRE. (Fig. 4) Calibration was excellent for both protein risk models (10-year model: model-

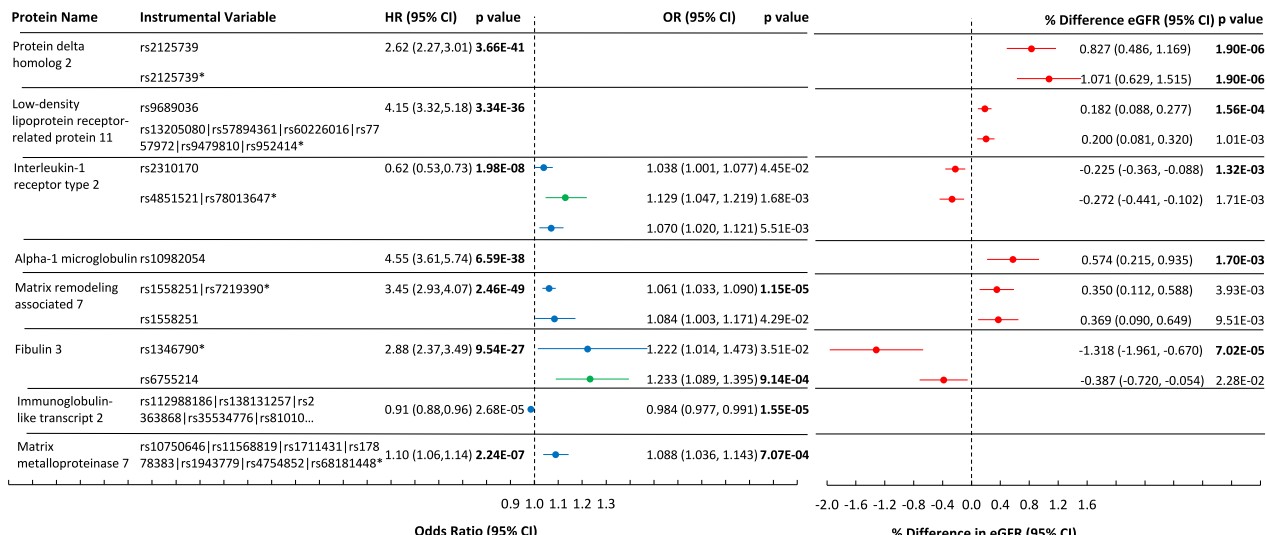

**Fig. 3 | Mendelian randomization of proteins associated with CKD progression.** Eight proteins were linked to pQTLs that had significant Mendelian randomization associations after correction for multiple tests in at least one GWAS for cross-sectional kidney function (eGFR) or for CKD progression (Rapid3 or CKDi25). HR (95% CI) per $\log_2$ of protein are shown, for the outcome of ESRD or 50% decline in eGFR, in CRIC, with adjustment for eGFR. pQTLs were identified in CRIC and in deCODE (deCODE marked with *). The three GWAS used are eGFR (red), Rapid3 (blue), and CKDi25 (green). Mendelian randomization associations are shown as red points (% difference in eGFR), or blue and green points that represent odds ratio. If an association met significance after multiple testing, *P* value is in bold. All *P* values are two-sided.

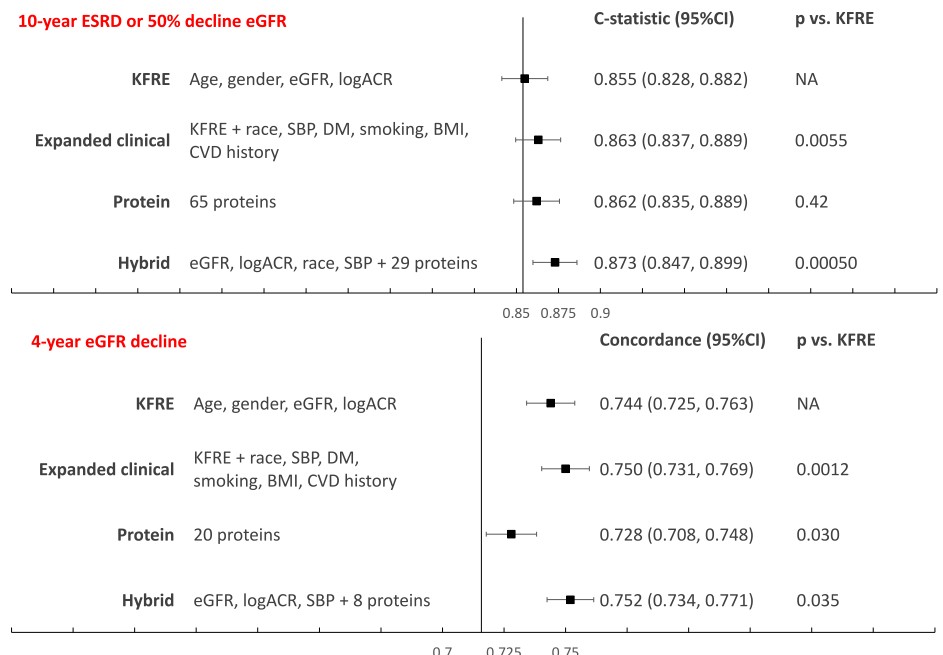

**Fig. 4 | Risk models for primary and secondary outcomes.** For the primary outcome, C-statistics and concordance are calculated in the CRIC testing set, $N = 577$ participants with 186 events. For eGFR slope, $N = 571$ participants, $P$ values calculated by two-sided concordance testing. Points represent C-statistic for primary outcome, and concordance for the secondary outcome. The whiskers stand for standard error. KFRE Kidney Failure Risk Equation, ACR albumin-to-creatine ratio, eGFR estimated glomerular filtration rate.

based calibration $P > 0.1$ for all except Q1, with only eight events; dichotomous slope calibration $P \geq 0.05$ for all except Q1) (Supplementary Fig. 2) Both protein models show a broad dynamic range of prediction, with the ratio of quintile 5/quintile 1 of predicted as well as observed risk being 20 for the primary outcome and 10 for the binary eGFR slope outcome. Through a search of the Druggable Target Database[15], 14 of the 65 proteins included in the risk model for the primary outcome, and 3 of 20 in the model for the secondary outcome are currently druggable targets (Supplementary Data 11 and 12).

### Sensitivity analyses

The 65-protein model for the primary outcome showed similar discrimination in subgroups of diabetes status, race, and eGFR ($P$ for interaction >0.1 for all) (Supplementary Fig. 3). We examined whether discrimination varied by the length of follow-up. Time-dependent AUCs for the 65-protein model and other risk models for follow-up periods ranging between 1 and 15 years show higher AUCs in the short term. (Supplementary Fig. 4). The hybrid model incrementally surpassed the KFRE at 5 years and 15 years: 5-year hybrid 0.90 (0.87, 0.93) vs KFRE 0.89 (0.86, 0.91) ($P = 0.01$); 15-year hybrid 0.87 (0.84, 0.89) vs KFRE 0.85 (0.82, 0.88) ($P = 0.0005$). We additionally explored creating distinct multiprotein risk models for different time horizons and found higher c-statistics for shorter time horizons for both KFRE and protein models (Supplementary Data 13).

We also evaluated the effect of calculating eGFR using a creatinine-based race-free equation on the performance of the risk models. Discrimination of the protein model, as well as for KFRE and hybrid models, for the primary outcome, was unchanged: protein model C-statistic (SE) 0.862 (0.014) in our initial analysis, 0.860 (0.014) using race-free equation. Discrimination of the protein model (and other models) for eGFR slope was slightly lower using the race-free eGFR: C-statistic (SE) 0.728 (0.010) in our initial analysis, 0.684 (0.010) using the race-free equation. (Supplementary Data 14) In additional sensitivity analyses, we evaluated whether KFRE with coefficients refit to our cohort performed better for the primary outcome than the original KFRE equation. In the 20% testing set, C-statistics were similar between the original KFRE[10] (0.844 (95% CI: 0.817, 0.872)), and the refit KFRE (0.855 (95% CI: 0.828, 0.882)). For the outcome of 10-year kidney failure alone (without the endpoint of 50% decline in eGFR), the C-statistics (95% CI) were also similar: original KFRE 0.892 (95% CI: 0.870, 0.915); refit KFRE 0.894 (95% CI: 0.872, 0.916).

### Validation of the 65-protein model in ARIC

The validation cohort was comprised of 578 ARIC participants with eGFR<60 ml/min/1.73 $m^2$ at ARIC Visit 3. The C-statistic (95% CI) for the 65-protein model in ARIC validation was 0.840 (0.785–0.896). The calibration of the 65-protein model in ARIC was fair (GND chi$^2$ = 9.2, $P = 0.06$) overall. Calibration was good in the highest two quintiles of risk but may have suffered from few events in the lowest three quintiles of predicted risk, potentially leading to discrepant predicted vs observed estimates (each of these 3 quintiles having <8 events). Calibration was good in the 4th and 5th quintiles of predicted risk, each having 12 and 57 outcome events, respectively (Supplementary Data 2).

## Discussion

In this study of proteomics of CKD progression, we quantified 4638 unique plasma proteins in 3249 participants of CRIC and validated our findings in 578 participants from ARIC with CKD, comprising a total of nearly 18 million individual protein measurements. We identified over 500 proteins associated with CKD progression after adjustment for eGFR and 100 proteins after extensive covariate adjustment at the Bonferroni-corrected statistical significance threshold. Individual protein and canonical pathway analyses highlight potential roles of ephrin signaling, BMP antagonists, and prothrombin activation. We identified 8 plasma proteins with potentially causal significant associations by MR; 5 of these have not been previously identified by MR, and 3 are currently druggable targets. Applying machine learning, we developed proteomic risk models for long- and short-term CKD progression with a similar excellent predictive utility to the refit KFRE clinical model but, in contrast to the refit KFRE, the protein models consist of modifiable risk factors[11].

Four of the top twenty proteins identified in CRIC, validated in ARIC, and associated with a higher risk of CKD progression are antagonists of BMPs, also known as growth differentiation factors (GDFs) (Table 1). BMPs were originally discovered as constituents of bone extract that cause ectopic bone formation when implanted in rats[16]. More than 30 BMPs form a subgroup of the transforming growth factor-β (TGF-β) superfamily, with diverse skeletal and extraskeletal functions[17]. BMP antagonists include Gremlin, sclerostin, follistatin, noggin, and brorin, and there is evidence that these antagonists play a role in modulating the extracellular matrix (ECM)[18]. There has been interest in BMP antagonists for treating renal disease: for example, Gremlin, an antagonist of BMP2 and 4, may be protective of diabetic nephropathy in experimental models[19]. TWSG1, a protein risk factor also found to be an independent risk factor for CKD progression[20], with a nominally significant MR association in CKDi25 in our study, is an antagonist of BMP 7, which is produced in the kidney and is protective against renal fibrosis and other types of renal injury in experimental models[21]. FSTL3 is another BMP antagonist previously shown to predict CKD progression[20], which we found to have a nominal ($P < 0.05$) association by MR in CKDi25 GWAS. FSTL3, a 30 kDa protein, is an antagonist of BMP2 and 4 (both of which promote bone formation and other processes), GDF8 (a growth factor for skeletal muscle), and GDF11 (a factor negatively associated with age-related left ventricular hypertrophy)[22–24]. FSTL3 is renally cleared, and its hepatic production may be increased in renal disease[25]. Thus, there is plausibility to our findings that members of the BMP family are involved in CKD progression.

Five of the top 20 proteins identified in CRIC and validated in ARIC associated with a higher risk of CKD progression are members of the Ephrin family (Table 1) and Ephrin signaling was among the top canonical pathways identified in our study (Table 2). Ephrin receptors interact with vascular endothelial growth factor to control angiogenesis[26], and CKD is characterized by microvascular disease and capillary rarefaction within the kidney. Ephrin type-B receptor 4 stimulates angiogenesis after kidney injury to enhance recovery[27]. Ephrin-B2 knockout mice are protected from renal fibrosis in a renal ischemia model, suggesting that ephrin-B2 facilitates renal fibrosis[28].

The prominence of the canonical pathway of prothrombin activation among proteins associated with CKD progression in our study might be explained by interactions of thrombin with protease-activated receptors that are found in several cell types in the kidney[29]. Thrombin may have direct effects on the kidney via protease-activated receptor 1 (PAR1), which is activated by thrombin and found in several different cell types. PAR1 deficiency is protective against diabetic nephropathy in animal models[30].

We also found several proteins that are associated with a lower risk of CKD progression (Supplementary Data 4 and 6). C1GALT1-specific chaperone 1 was associated with a lower risk of the primary and secondary outcome of CKD progression in CRIC and passed the criteria for validation in ARIC for the primary outcome. This protein facilitates protein glycosylation and platelet activation. CILP2 was associated with a lower risk of CKD progression in CRIC and ARIC and had a nominally significant MR association ($P < 0.05$). CILP1 levels are increased in the myocardium after infarction, and CILP1 is thought to protect against fibrosis in the myocardium by inhibiting TGFβ[31]. CILP2 could have a similar anti-fibrotic effect in the kidney.

Our MR analysis revealed eight potentially causal mediators of CKD progression that were significant after adjustment for multiple tests in one or more renal GWAS. Five of these proteins, to our knowledge, have not been shown previously to have MR associations: EGFL9 is an antagonist of the NOTCH pathway[32] which has roles in kidney development and disease[33]. LRP-11 is a membrane protein related to lipid metabolism. MXRA7 is an extracellular matrix protein. IL-1 sRII and ILT-2 are immunologic receptors, and both are currently druggable targets.

Given that approximately half of CRIC participants have diabetes mellitus, our study provides an opportunity for characterizing differences in proteins that predict the progression of diabetic vs. non-diabetic kidney disease. We validated several proteins previously found to predict higher or lower risk of kidney failure among individuals with CKD all of whom had diabetes[12,13] (Supplementary Information Fig. 1 and Supplementary Data 8). Yet, overall, we found that many proteins predict CKD progression similarly, irrespective of diabetes status, suggesting shared mechanisms of progression of diabetic and nondiabetic CKD. However, two proteins did have significantly different statistical associations among patients with diabetes compared to those without. Interleukin-18 receptor 1 predicted higher risk and angiopoietin-1 predicted lower risk among patients with diabetes. Stratification by diabetes may be an important component for the future discovery of biomarkers of CKD progression, with the expectation that while few markers may differ by diabetes status, these differences could be important for developing therapeutics for different etiologies of kidney disease.

The KFRE equation was developed to predict kidney failure over 5 years and has shown excellent validation in meta-analyses of international studies[10]. It is accessible to clinicians, given that the four factors of age, gender, eGFR, and albuminuria can be readily determined. A key limitation of the KFRE is that it sheds little light on the biological mechanisms by which CKD progresses in individual patients and besides albuminuria, its components are not readily modifiable. Plasma levels of proteins readily change in response to lifestyle and pharmacological interventions. The 65-protein model derived in this study for a 10-year 50% decline in eGFR or kidney failure matched the KFRE for its excellent discrimination and had even better discrimination at 5 years. A separately derived protein model for kidney failure alone at 2 years had a C-statistic of 0.95 (similar to the KFRE applied to 2 years, 0.94). Short-term protein models could be used as surrogate outcomes in clinical trials of therapeutics. Clinicians might use the protein model not only to identify patients at higher risk of kidney failure, but also to monitor patients' response to lifestyle and medication changes. Showing the patient that his or her risk score has improved could improve compliance with medications. Hybrid clinical-protein models showed modest statistically significant improvement over KFRE. Notably, the addition of clinical factors added little to the discrimination of protein models. One may conclude from this that proteins encode demographic and clinical information in addition to carrying important biological signals, a concept that we have demonstrated previously[11].

Our study has numerous strengths, but we also acknowledge limitations. Additional clinical and experimental approaches informed by our proteomics findings will be needed to establish conclusively which of the protein biomarkers identified in our study are involved as causal mediators in CKD progression, given the limits of epidemiological association studies. The prognostic utility of the multiprotein risk score, and its capacity to reflect effects of medications, could be validated using samples from clinical trials involving kidney endpoints. The biological roles of specific proteins could be elucidated with animal models. While the CRIC population is well-phenotyped and affords extensive multivariable adjustment, any unmeasured confounders may bias the assessments of individual proteins as independent risk markers. We measured circulating and not tissue proteins, since plasma is more readily accessible as a diagnostic matrix than kidney biopsy tissue. Future studies are expected to correlate proteomic information from plasma and kidney biopsies. Lastly, the present Mendelian randomization analyses may be augmented by utilizing a more comprehensive GWAS for renal function that includes a meta-analysis of CKD Genetics Consortium and UK Biobank[34].

In conclusion, we present the largest proteomic study of participants with CKD to date with a total of nearly 18 million individual protein measurements, in a well-phenotyped population of >3000

participants. Our analyses reveal multiple individual protein risk factors for CKD progression that have not been previously described, and we show that individual proteins and a 65-protein risk model for 10-year CKD progression replicate well in ARIC. Druggable targets within our protein risk models and significant MR findings may provide the impetus for developing therapeutics. Biological pathways and individual proteins that we have identified, including BMP antagonists, ephrin signaling, and prothrombin activation warrant further study.

## Methods

### Participants

The CRIC study protocols adhered to ethics regulations of each institution where participants were enrolled, requiring approval from the following committees: University of Pennsylvania Institutional Review Board, Federalwide Assurance # 00004028; Johns Hopkins Institutional Review Board NA_00044034/CIR00004697; The University of Maryland, Baltimore Institutional Review Board; University Hospitals Cleveland Medical Center Institutional Review Board; MetroHealth Institutional Review Board; Cleveland Clinic Foundation Institutional Review Board IRB #5969; University of Michigan Medical School Institutional Review Board; Wayne State University Institutional Review Board; University of Illinois at Chicago Institutional Review Board; Tulane Human Research Protection Office, Institutional Review Boards, Biomedical Social Behavioral, IRB #140987; Kaiser Permanente Northern California Institutional Review Board. The Atherosclerosis Risk in Communities (ARIC) Study adhered to ethics regulations from and was approved by a single Institutional Review Board (sIRB) at Johns Hopkins School of Medicine (FWA00005752; IRB00311861) and Institutional Review Boards (IRB) at all participating institutions: University of North Carolina at Chapel Hill, Johns Hopkins University School of Public Health, University of Minnesota, Wake Forest University Health Sciences, University of Mississippi Medical Center, Baylor College of Medicine, University of Texas Houston Health Science Center, and Brigham and Women's Hospital. Study participants provided written informed consent at all study visits.

The CRIC study was designed to investigate risk factors for progression of CKD, incident cardiovascular disease, and overall mortality in persons with CKD[14]. Between 2003 and 2008, the CRIC study enrolled a total of 3939 ethnically diverse men and women at 7 clinical centers, ages 21–74 years, with eGFR 20–70 ml/min/1.73 m² by the simplified Modification of Diet in Renal Disease equation[14]. Eligibility criteria and baseline characteristics of the CRIC cohort have been published[14,35]. The CRIC study was approved by the Institutional Review Boards of the participating centers, and the research was conducted in accordance with the principles of the Declaration of Helsinki. All study participants provided written informed consent. At enrollment, information on participant sex/gender was collected by self-report; there were no sex/gender-based inclusion or exclusion criteria. For the present analysis, plasma samples from 3419 CRIC participants from the year 1 visit, considered our study's baseline, were assayed with SomaScan V4.0. Each sample was assayed once with SomaScan. We excluded 53 participants with prevalent kidney failure. Due to the interference of lupus antibodies with aptamers (communication from SomaLogic), we also excluded 12 participants with systemic lupus erythematosus. After 105 samples were excluded that did not pass SomaLogic's quality control standards, the final analytical cohort consisted of 3249 participants. Fourteen participants were excluded who did not have a baseline measure of eGFR, leaving 3235 individuals eligible for analyses of the primary outcome of a 50% decline in eGFR or kidney failure over 10 years and 3243 participants eligible for analyses for the secondary outcome of a 4-year eGFR decline.

### SomaScan version 4.0

SomaScan is an assay based on modified aptamers, which are chemically modified single strands of deoxyribonucleic acid ~40 nucleotides long, as binding reagents for target proteins[7,8,36–39]. Modified aptamers bind to proteins with high affinity similar to antibodies (lower limit of detection $10^{-15}$ moles per liter)[36–38] "Pull-down" studies, in which the aptamer-protein complexes were isolated and the identities of the bound proteins were verified by targeted mass spectrometry and gel electrophoresis, have been performed for 920 proteins among 1305 proteins in a previous version of the assay[39]. These studies showed that >95% of aptamers correctly targeted the intended proteins (for those proteins in concentrations sufficient to be detected by mass spectrometry). The samples on the SomaScan assay are run at three different dilutions to assay each analyte within its linear range of concentrations. The assay results are quantified on a hybridization microarray and reported in RFU. SomaLogic has procedures for data calibration, standardization and internal controls, typical of microarray technologies.

The SomaScan V4.0 menu includes 5284 aptamers (Supplementary Data 15). We excluded 305 aptamers paired to non-human proteins, 130 incompletely characterized investigational aptamers, and 19 aptamers with >50% coefficients of variation (CVs) in 129 split duplicates from CRIC participants that were run simultaneously to our large-scale proteomic study. This left 4830 aptamers and 4638 unique proteins (some proteins are measured by 2 or more aptamers) (Supplementary Data 16). The median intra-assay CVs, from plasma of healthy individuals are reported as ≤5%[12,40]. We conducted our quality control study using split duplicate plasma samples from CRIC participants with CKD stages 3A, 3B, and 4. Median split duplicate CVs were ≤5% and did not vary by the stage of CKD or by diabetes status[41].

### Study outcomes

The primary outcome was time to the first of two clinical outcomes, i.e., ≥50% eGFR decline or incident kidney failure (defined as the need for renal replacement therapy), within a 10-year time horizon. To capture short-term CKD progression, we analyzed the 4-year eGFR slope as a secondary outcome, generated using a linear mixed effect model with a random intercept and a random slope. The eGFR slope was formulated as a continuous variable, and alternatively as a dichotomized endpoint of eGFR decline ≥ versus < 3 mL/min/1.73 m² per year. The slope was censored at kidney failure. For the derivation of risk models, we wished to optimize the accuracy of eGFR measures specifically among CRIC participants, and for this reason GFR was estimated using the 5-variable CRIC equation including serum creatinine, serum cystatin, age, gender, and race, given this equation has been extensively validated among CRIC participants as the closest estimate of GFR measured by iothalamate clearance[42]. In sensitivity analyses, we estimated GFR using the 2021 CKD EPI creatinine that is based on age, sex, and creatinine and omits race as a variable[43,44].

### Covariate definitions

Study covariates were chosen a priori based on the literature and used definitions published by CRIC[45]. Diabetes mellitus was defined by a fasting glucose of ≥126 mg/dL or the use of insulin or oral hypoglycemic medications. Hypertension was defined by systolic blood pressure ≥140 mm Hg, diastolic blood pressure ≥90 mm Hg, or the use of antihypertensive medications. Lifestyle, sociodemographic and medical history information was obtained at baseline using self-reported questionnaires, including gender, race, ethnicity, and smoking status. Prevalent cardiovascular disease at entry was assessed by a self-reported history of prior myocardial infarction, coronary revascularization, heart failure, stroke, or peripheral artery disease. Body mass index was calculated using measured height and weight and expressed in kilograms per meter squared. At the visit with proteomics, albuminuria was not directly measured. Albuminuria was calculated from urine protein to creatinine ratio using the crude (unadjusted) equation as published in ref. 46.

## Statistical analysis

Summary statistics for the CRIC participants' baseline characteristics were calculated as mean and standard deviation (SD) for symmetric variables and median and interquartile range (IQR) for skewed variables. SomaLogic normalizes the entire protein dataset using Adaptive Normalization by Maximum Likelihood (ANML) to remove unwanted biases in the assay. ANML is an iterative procedure that adjusts values for analytes that fall outside expected measurements from a reference distribution. Protein values are reported in relative fluorescent units (RFU) after ANML normalization. We chose to standardize RFU using median absolute deviation (MAD); this approach allows for the ranking of predictors and is more robust than conventional methods of standardization (mean subtraction, standard deviation division) for skewed data. We Winsorized (clipped) outliers at median ± 5 MAD.

The Cox proportional hazards regression model was used to assess the association between individual proteins and the primary outcome. Associations of individual proteins with continuous eGFR slope were assessed using multivariable linear regression. In each instance, we constructed models with three levels of adjustment: (i) no adjustment, (ii) adjustment for eGFR only, or (iii) adjustment for age, gender, race, eGFR, log[urine protein to creatinine ratio], systolic blood pressure, diabetes, smoking status, body mass index, and cardiovascular disease history. Evaluating each individual protein was a preliminary step, prior to determining which proteins to replicate externally, and then to examine with Mendelian randomization. In order to rank individual proteins by strength of association with the outcome, we employed MAD standardization because it is more robust than $\log_2$ standardization for skewed predictors. We chose to select "top hits" from among the protein associations meeting a significance threshold of FDR < 0.05, rather than Bonferroni significance, to minimize type II error at the screening stage. The Benjamini–Hochberg (BH) method was used to control the false discovery rate (FDR) at 5%[47,48]. We then selected protein "top hits" by effect size per MAD unit. We present these top hits in tables using HR per $\log_2$ to illustrate effect sizes on a scale more commonly used in epidemiology than MAD. Presentation tables also include the $P$ value in order to illustrate that most of these proteins meet the Bonferroni-corrected statistical significance level ($P < 1.0 \times 10^{-5}$ after adjusting for ~5000 tests).

To determine whether any associations of protein biomarkers with CKD progression may be unique to people with diabetes, we explored the impact of diabetes on associations of individual proteins with the primary outcome by visualizing a scatterplot of HRs in participants with vs. without diabetes. We also performed formal statistical interaction testing by diabetes status for all proteins that were associated with the primary outcome. This analysis included the 17 KRIS proteins reported to predict kidney failure in patients with diabetes[12] and three additional proteins reported by the same investigators as potentially protective of kidney failure in patients with diabetes[13].

We developed multiprotein risk models for the prediction of CKD progression and compared their predictive performance to clinical and hybrid clinical-protein models. We randomly split the CRIC data into two sets: 80% of individuals comprised the training set, and the remaining 20% the testing set. We used the training set to build prediction models and determine attendant tuning parameters. The testing set was used solely to evaluate the models' performance. Our frontline technique for developing protein risk prediction models was elastic-net (EN) Cox regression which combines ridge (L2) and LASSO (L1) penalties and handles each of the three (time-to-event, continuous, binary) outcome types. Model fitting was conducted using the R package *glmnet*[11,12]. The relative contributions of the two penalties are controlled by a mixing parameter $\alpha$ which we set to 0.5 for balance. The shrinkage (regularization) parameter $\lambda$ which controls model complexity (the number of included proteins) was determined by tenfold cross-validation and the "1 standard error rule". After the final selection of proteins, to reduce bias in estimated regression coefficients[49], we refit

the selected features for the EN model in a Cox regression model for the CKD progression survival outcomes and a logistic regression model for the binary eGFR decline outcome, as previously published[50].

We evaluated predictive performance by calculating Harrell's C-index[47] or Receiver Operating Characteristics (ROC) Area under the Curve (AUC) in the testing set[47]. For survival outcomes, we additionally calculated time-dependent AUC for years 1 to year 15 using the testing set data. We evaluated model calibration in the training set with calibration bar plots to visualize the agreement between predicted and observed risk in each quintile of participants defined by predicted risk. A formal assessment of calibration made recourse to a model-based test that can accommodate survival endpoints in addition to continuous and binary outcomes[51]. We further conducted stability analyses of our EN models to ensure that results were not overly dependent on the specific training / test set partition deployed. This involved repeating the entire EN procedure on five alternate random partitions into training and test sets.

We compared the protein models for CKD progression to two clinical risk models. The first model was comprised of variables from the 4-variable Kidney Failure Risk Equation (KFRE) model (age, gender, eGFR, urine albumin-to-creatine ratio)[10]. We also used a 10-variable clinical model (referred to herein as an expanded clinical model) that included the 4 KFRE variables, plus 6 other variables reported to associate with CKD progression in CRIC (race, systolic blood pressure, diabetes, smoking status, BMI, and cardiovascular disease (CVD) history). To optimize the performance of these clinical models in CRIC, the coefficients of the variables of both clinical models were refit to the primary and secondary outcomes. Comparisons between the various risk models were based on C-statistics calculated in the same participant set, using significance as a two-sided $P$ value < 0.05, and visualized using forest plots.

In sensitivity analyses, we examined discrimination of the 65-protein risk model for the primary outcome in subgroups of gender, race, diabetes or eGFR. Furthermore, in addition to the 10-year time horizon for risk modeling, we evaluated the performance of other protein models derived for shorter or longer time horizons of 2, 5, and 15 years. We also evaluated the performance of our primary 65-protein model protein model using a race-free creatinine-based equation to calculate a 50% eGFR decline for the primary outcome.

## External validation

We performed external validation for the primary outcome in 578 participants at visit 3 of the ARIC Study[52] who had CKD (eGFR < 60 ml/min/1.73 m$^2$) when plasma was obtained for SomaScan V4.0 proteomic analysis. We performed validation of 20 individual proteins with the highest and 20 proteins with the lowest HRs for the primary outcome in CRIC after adjustment for eGFR, by performing Cox regression for their associations with the same outcome in ARIC, with adjustment for eGFR. The statistical criterion for validation was a Bonferroni $P$ value of <(0.05/40) or <0.00125, based on correcting for 40 proteins carried forward for validation. Discrimination and calibration of the multiprotein model for the primary outcome from CRIC were tested in ARIC, the calibration after adjustment for differences in baseline hazard, but retaining coefficients developed in CRIC. Statistical analyses were performed using R, version 4.0.3 (RStudio, Inc., Boston, MA. URL http://www.rstudio.com/), with the packages of *glmnet* (version 4.0-2), *survival* (version 3.2-7 *pec* (version 2019.11), *compareC* (version 1.3.1), *forestplot* (version 1.10), lme4 (version 1.1-26).

## Pathway analysis

We performed pathway analyses to elucidate the biological processes and regulatory mechanisms associated with CKD progression. The set of CKD progression-associated proteins with HRs significant at a false discovery rate (FDR) threshold of 0.05 after adjustment for eGFR were organized into canonical pathways by the Ingenuity

Pathway Analysis (IPA) tool as we have described previously[6,9,53,54]. For those modified aptamers that had multiple Uniprot identifications associated with 1 result, only the first Uniprot identification listed was used[6,9,53]. For proteins measured by two or more aptamers, the aptamer measurement with the largest effect size was utilized for the analysis. The Fisher right-tailed exact test was used to calculate a *P* value to determine the probability that the association of the differently expressed proteins in the measured dataset, and the pathway are explained by chance alone.

## Mendelian randomization

To investigate the potential causality of CKD progression for a limited set of proteins from our study, we conducted Mendelian Randomization (MR) analysis for 76 aptamers (75 proteins) that were either discovered as risk factors for CKD progression in CRIC and successfully validated in ARIC or were included in the 65-protein risk model for the primary outcome in CRIC. Genotyping has been performed in CRIC using Illumina HumanOmni1-QUAD V1.0[55] with 7,102,205 measured or imputed genetic variants available for pQTL analysis (861,291 variants prior to imputation). For each protein, we performed the protein quantitative loci (pQTL) analysis and considered *cis*-pQTL variants within 1 megabase (Mb) upstream or downstream of the transcription start site of the corresponding protein-coding gene that had a *P* value < 5e-6. Furthermore, we conducted the conditional association analysis within the candidate set with the GCTA-COJO software[56] and selected the conditional significant variants with the p-value threshold $5 \times 10^{-6}$ for the subsequent MR analysis. pQTL-protein associations were adjusted for age, gender, eGFR, BMI, and the first five genotype principal components. For proteins with more than one single nucleotide polymorphism (SNP) selected, we used multi-SNP MR using the inverse variance weighting method[57]. For proteins with just one SNP selected as the instrumental variable, we estimated the causal effect using the Wald ratio test[58]. R packages Mendelian Randomization[59] and TwoSampleMR[58] were used in our analyses. Since most genome-wide association studies (GWAS) of CKD focus on European Ancestry (EA), we restricted our pQTL analysis for this study to 1208 CRIC participants of European Ancestry. We augmented our MR analysis using published significant pQTLs for the SomaScan V4.0 in the deCODE, a cohort of 35,559 Icelandic participants, for which the methods have been previously published[60]. Utilizing *cis* pQTLs from CRIC or deCODE, we searched within three publicly available GWAS for kidney function to determine whether these variants were associated with kidney function decline, designating the significance threshold as *P* value = 0.05/# distinct proteins queried in the GWAS. We chose three publicly available GWAS datasets assembled from the CKD Genetics Consortium and the United Kingdom Biobank. The eGFR dataset includes 567,460 participants of European descent with eGFR measures within the CKD Genetics Consortium[61]. Rapid3 and CKDi25 include 42 cohorts from either CKD Genetics Consortium or UK Biobank with serial kidney function measures[62]. Rapid3 includes 34,874 cases in whom eGFR decline was ≥3 ml/min/1.73 m² and 107,090 controls. CKDi25 includes 19,901 cases who start at eGFR >60 ml/min/1.73 m² and decline to less than 60 ml/min/1.73 m² and have ≥25% decline in eGFR, as well as 175,244 controls[62]. GWAS are available at http://ckdgen.imbi.uni-freiburg.de/.

## Reporting summary

Further information on research design is available in the Nature Portfolio Reporting Summary linked to this article.

## Data availability

The CRIC data are available from the CRIC Study group upon request and with a Data Use Agreement. Data requests can be made by contacting the CRIC Scientific and Data Coordinating Center at cri-projmgmt@lists.upenn.edu. Data access is controlled due to the terms of the informed consent which does not allow for the data to be posted publicly with open access. The CRIC Study group will typically respond to requests within one week, but the timeframe for providing data will vary depending on the timeline for completion of a Data Use Agreement between Penn and the receiving institution. Among other standard terms, Data Use Agreements will stipulate that the data only be used for a pre-specified purpose and that the data not be shared with third parties, and that publications of the data must appropriately recognize the source of the data. If at the time of the data request, the data are available through a federal repository, the requestor will be referred to the appropriate repository to submit a data request. ARIC data are regularly posted to repositories, including dbGaP and Bio-LINCC in addition, requests for data and verification analysis can be sent to the ARIC data coordinating center at University of North Carolina (aricpub@unc.edu attention David Couper) Source data are provided with this paper. Please see Supplementary Data Files and Source Data, Fig. 2, for minimum datasets that delineate protein associations described herein. Source data are provided with this paper.

## Code availability

R code for these analyses may be requested from the CRIC Scientific and Data Coordinating Center at cri-projmgmt@lists.upenn.edu.

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

## Acknowledgements

We thank CRIC Study Investigators, including Lawrence J. Appel, MD, MPH; Debbie L. Cohen, MD; Robert G. Nelson, MD, PhD, MS; Panduranga S. Rao, MD; Vallabh O. Shah, PhD, MS; Mark L. Unruh, MD, MS. Grants received by individual authors: R01GM129781 (to H.L.). R01DK124399 (to M.E.G. and J.C.). Funding for this study was obtained under a cooperative agreement from the National Institute of Diabetes and Digestive and Kidney disease (5U01DK108809 to P.G., R.F.D., and R.D.). Funding for the CRIC Study was obtained under a cooperative agreement from National Institute of Diabetes and Digestive and Kidney Diseases (U01DK060990, U01DK060984, U01DK061022, U01DK061021, U01DK061028, U01DK060980, U01DK060963, U01DK060902 and U24DK060990). In addition, this work was supported in part by: the Perelman School of Medicine at the University of Pennsylvania Clinical and Translational Science Award NIH/NCATS UL1TR000003, Johns Hopkins University UL1 TR-000424, University of Maryland GCRC M01 RR-16500, Clinical and Translational Science Collaborative of Cleveland, UL1TR000439 from the National Center for Advancing Translational Sciences (NCATS) component of the National Institutes of Health and NIH roadmap for Medical Research, Michigan Institute for Clinical and Health Research (MICHR) UL1TR000433, University of Illinois at Chicago CTSA UL1RR029879, Tulane COBRE for Clinical and Translational Research in Cardiometabolic Diseases P20 GM109036, Kaiser Permanente NIH/NCRR UCSF-CTSI UL1 RR-024131, Department of Internal Medicine, University of New Mexico School of Medicine Albuquerque, NM R01DK119199. The ARIC Study is carried out as a collaborative study supported by National Heart, Lung, and Blood Institute contracts (HHSN268201700001I, HHSN268201700002I, HHSN268201700003I, HHSN268201700004I, and HHSN268201700005I) and Neurocognitive 2U01HL096812, 2U01HL096814, 2U01HL096899, 2U01HL096902, and 2U01HL096917 from the NIH (NHLBI, NINDS, NIA, and NIDCD). SomaLogic Inc. conducted the SomaScan assays for ARIC in exchange for use of ARIC data. We thank the staff and participants of the ARIC study for their important contributions. The opinions expressed in this paper do not necessarily reflect those of the National Institute of Diabetes Digestive and Kidney Disease, the National Institutes of Health, the Department of Health and Human Services or the Government of the United States of America.

## Author contributions

R.F.D. conceived and obtained funding for the project, conceived the study design, supervised analyses, drafted and revised the manuscript. R.D. assisted with obtaining funding, overall study design, performed IPA analyses, and reviewed the manuscript. Y.R. performed all proteomics analyses, including machine learning, devising risk models, creating figures and tables for presentation. J.W. performed Mendelian randomization analyses. Z.Z. contributed to proteomics analyses. H.S. reviewed the manuscript. A.G. reviewed the manuscript. A.P. advised on eGFR slope analyses and critically reviewed the manuscript. J.P.L. reviewed the manuscript. M.R. reviewed the manuscript. C.H. advised on the application of the clinical risk models in CRIC and reviewed the manuscript. M.R.W. reviewed the manuscript. J. Chen reviewed the manuscript. A.A. advised on eGFR slope analyses and critically reviewed the manuscript. M.E.G. supervised A.S. and reviewed the manuscript. A.S. performed validation analyses in ARIC. J.C. advised on validation analyses in ARIC and reviewed the manuscript. H.L. supervised Y.R. P.L.K. reviewed the manuscript. R.S.V. reviewed the manuscript. H.F. assisted with study design, advised on all aspects related to the CRIC cohort, and reviewed the manuscript. M.R.S. advised on study design and supervised Y.R. P.G. conceived and obtained funding for the project, conceived the study design, supervised analyses, reviewed and revised the manuscript.

## Competing interests

J. Coresh is a scientific advisor receiving fees from SomaLogic and Healthy.io. P.G. serves on the advisory board of SomaLogic Inc., but accepts no salary, honoraria, or any other financial incentives. M.R.W serves as a consultant for AstraZeneca, Bayer, CSL Vifor, Boehringer-Ingelheim, Johnson & Johnson, CareDx. C.H. is the principal investigator of an investigator-initiated grant proposal with Satellite Healthcare, receives payment as author for UpToDate, provides expert opinion for Triangle Insights Group, Aria Pharma, LG Chem, and legal consulting for Allen Shepherd & Lewis, King and Spalding, Lewis Brisbois, McMasters Keith Butler. The remaining authors declare no competing interests.

## Additional information

[1]Division of Nephrology, University of Texas Southwestern Medical Center, Dallas, TX, USA. [2]Division of Cardiovascular Medicine, Perelman School of Medicine at the University of Pennsylvania, Philadelphia, PA, USA. [3]Department of Biostatistics, Epidemiology, and Informatics, Perelman School of Medicine, University of Pennsylvania, Philadelphia, PA, USA. [4]Harvard T.H. Chan School of Public Health, Boston, MA, USA. [5]Division of Research, Kaiser Permanente Northern California, Oakland, the Department of Health Systems Science, Oakland, CA, USA. [6]National Institute of Diabetes and Digestive and Kidney Diseases, National Institutes of Health, Bethesda, MD, USA. [7]Department of Medicine, University of Illinois Chicago, Chicago, IL, USA. [8]Department of Medicine, University Hospitals Cleveland Medical Center, Case Western Reserve University School of Medicine, Cleveland, OH, USA. [9]Division of Nephrology, University of California San Francisco, San Francisco, CA, USA. [10]Division of Nephrology, Department of Medicine, University of Maryland School of Medicine, Baltimore, MD, USA. [11]Department of Epidemiology, Tulane University, New Orleans, LA, USA. [12]Welch Center for Prevention, Epidemiology, and Clinical Research, Johns Hopkins University, Baltimore, MD, USA. [13]Department of Medicine, Johns Hopkins University, Baltimore, MD, USA. [14]Division of Precision Medicine, New York University Grossman School of Medicine, New York, NY, USA. [15]Division of Kidney, Urologic, and Hematologic Diseases, National Institute of Diabetes and Digestive and Kidney Diseases, National Institutes of Health, Bethesda, MD, USA. [16]University of Texas School of Public Health San Antonio and the University of Texas Health Sciences Center in San Antonio. Section of Preventive Medicine and Epidemiology, Department of Medicine, Boston University School of Medicine, Boston, MA, USA. [17]Department of Epidemiology and Biostatistics, University of California, San Francisco, San Francisco, CA, USA. [18]Division of Cardiology, University of California, San Francisco, San Francisco, CA, USA. [19]These authors contributed equally: Ruth F. Dubin, Rajat Deo. ✉e-mail: ruth.dubin@utsouthwestern.edu

## CRIC Study Investigators

**Mahboob Rahman[8], James P. Lash[7], Jing Chen[11], Alan S. Go[5], Chi-yuan Hsu[5,9] & Amanda Anderson[11]**

## CKD Biomarkers Consortium

**Ramachandran S. Vasan[16], Josef Coresh ®[12,13], Morgan E. Grams ®[12,13,14], Paul L. Kimmel[15], Ruth F. Dubin ®[1,19] ✉, Rajat Deo[2,19], Harold Feldman[3], Amanda Anderson[11], Haochang Shou ®[3] & Peter Ganz ®[18]**

A full list of members and their affiliations appears in the Supplementary Information.

