## [Peer Review File · Nature Communications]

Proteomics of CKD Progression in the Chronic Renal Insufficiency CohortREVIEWER COMMENTS

Reviewer #1 (Remarks to the Author):

Thank you for the opportunity to review the manuscript by Dubin et al..

The article titled "Proteomics of CKD Progression in the Chronic Renal Insufficiency Cohort" presents a comprehensive analysis of plasma proteins associated with the progression of chronic kidney disease (CKD) and identifies potential therapeutic targets for slowing CKD progression.

The study employs a large cohort of participants, an external validation and utilizes proteomic analysis techniques to explore the associations between plasma proteins and CKD progression. Overall, the article is well-written and provides very valuable insights into the molecular mechanisms underlying CKD progression. The findings of this study have the potential to advance our understanding of CKD progression and inform future research and therapeutic interventions in this field. However, there are a few points that might help to strengthen the study.

Major:

1. The introduction and discussion emphasize the need for a prognostic equation that includes modifiable factors and lists this as a limitation of the KFRE equation. I fully agree that both - good prognostic equations and the identification of modifiable / treatable factors – are very important challenges that require our attention. However, it is not clear to me why these need to be addressed in the same equation. Some clarification on that aspect or a rephrasing might be helpful.
2. Furthermore, the druggability was followed up on for the elastic net selected proteins. Elastic nets are not designed to identify all putatively causal variables and will for example pick one of two highly correlated variables (to a certain degree). A more conclusive list for potential downstream studies would be the same protein selection used for the MR screen.
3. A list of all analyzed proteins with some basic annotations (IDs, full names, etc) would be helpful supplementary material. Further the CVs from the 129 duplicates could be a nice resource to other SomaScan studies. This as well as the existing supplementary table 1 would benefit from a non-pdf format like xlsx.
4. Protein level transformation: The parallel presentation of log₂ and MAD unit analyses was

slightly confusing to me. E.g. the methods mention that HRs are given in log₂ units (implying only in log₂). Consider presenting log₂ and MAD unit HR in merged tables to ease the comparison and condense the supplementary materials. The actual MADs would also be a great addition to the proposed added supplementary table that lists all proteins. More importantly, MAD scaling and a log₂-transformation have different impacts on the distribution of proteins. What is the impact of the two transformations on skewness and similar parameters of the protein distributions? If robustness was the major concern driving the decision for MAD scaling, did you consider a rank-based inverse normal transformation?

5. Refitting of the EN model: “After the final selection of proteins, to reduce bias in estimated regression coefficients, we refit the selected features for the EN model ...”

Wouldn't the refitting after the variable selection introduce bias / overfitting instead of removing it? Isn't the penalty in the elastic net approach designed to limit the potential overfitting due to the high number of variables?

6. Significance testing:

a. The methods for the primary screen in CRIC lists both a Bonferroni threshold as well as a BH adjustment. For external validation you focused on the highest and lowest 20 HRs. Furthermore, some proteins that “narrowly missed” significance were reported. Results and discussion switch between FDR and Bonferroni reporting. In combination with three different adjustment levels this is challenging to follow. A more aligned multiple testing and validation design would be desirable.

b. Methods: “Bonferroni-corrected statistical significance levels were reported after adjusting for 4638 comparisons among proteins resulting in a nominal significance level of $p < 1.0 \times 10^{-5}$.” Technically it is 4830 comparisons, right? I agree with the choice to account for the number of proteins. Why is the threshold called nominal when it reflects a full Bonferroni correction? Did you apply the rounded threshold or $0.05/4638$?

c. The primary analysis on individual proteins focused the reporting on the fully adjusted models. The pathway analysis starts out with the proteins associated in the eGFR-only-adjusted models. Is there a specific reason driving this switch?

Minor:

- The abstract mentions the discrimination of the 65 protein model without including the actual statistic. Consider including a key discriminator.

- Abstract: “We found 100 plasma proteins independently associated with the primary CKD progression outcome.” Here, the independently would imply to me that all 100 proteins are significant in a joint model. However, I believe this to be referring to the 100 individual protein analyses that were significant. Consider removing “independently”.
- Consider switching nomenclature from ESRD to kidney failure according to KDIGO guidelines.
- Methods – Participants: Add “prevalent” in the following? “We excluded 53 participants with ESRD.”
- Methods – Study Outcomes: Why was the 2021 CKD EPI eGFR_{crea} equation instead of the 2021 CKD EPI eGFR_{crea-cys} equation used for sensitivity analyses? The second seems closer to the CRIC equation.
- Methods: “To optimize the performance of these clinical models in CRIC, the coefficients of the variables of both clinical models were refit...” Was this done using the same test dataset as the elastic nets were trained on?
- How were the conditional cis-pQTL analyses conducted? And how were the CRIC and deCODE pQTL combined in the MR analysis?
- For the eGFR MR a larger EA GWAS is available (N=1 million) from the CKDGen consortium (Stanzick et al. 2021; <https://www.nature.com/articles/s41467-021-24491-0>)
- Something went wrong in the formatting of references 34 and 35.

Reviewer #2 (Remarks to the Author):

In this manuscript, the author analyzed 3235 participants of CRIC study. They found 100 proteins which are independently associated with CKD progression outcome. These proteins were found to be associated with several biological pathways, including bone morphogenetic proteins, ephrin signaling, and prothrombin activation. The authors also developed multi-protein risk models for predicting CKD progression using machine learning and identified potential druggable targets. The study's methodology and experiments with a large number of cohort participants are carefully done and support the author's conclusions. However, the following concerns should be addressed before its publication at Nature communications.

Major comments

- The authors analyzed plasma samples from participants and developed a multi-protein risk model to predict CKD progression. However, it is unclear whether the changes in protein expression of plasma samples are truly linked to CKD progression that occurs in kidney tissue. Although the authors acknowledged this limitation in the Discussion section, they need to provide a more detailed plan to validate the proteomics information presented in their study.

- In the method section, the authors explained very well regarding participants in this study, statistical analysis, and derivation of risk models. However, the method used for detecting protein expression in plasma using SomaScan Version 4.0 was not well explained. As proteome analysis is a critical part of this study, it is recommended to provide more details about SomaScan. Specifically, the authors could explain the sample preparation process, the equipment used in the study, and provide a simplified explanation of how SomaScan works to detect proteins. This additional information would help readers to better understand and replicate the proteomics analysis.

- Regarding Figure 2, because protein name in volcano plot is quite small and its low resolution of the graph, it is tough to understand easily for the readers. In Supplementary Figure 1, protein name is also quite small.

- The authors identified druggable targets among the proteins associated with CKD progression and provided information about the drug names in the Supplementary Tables, including Tables 12 and 13. In order to facilitate their clinical translation in the future, it would be useful to provide additional information about these drugs, such as whether they have already been approved for other diseases or are still in the development phase.

- Related to the comment above, the authors state that there is correlation between plasma samples protein and CKD progression, and it might be a biomarker or druggable target. Is it applicable to the clinic? Please provide the potential strategy for detecting or preventing CKD progression.

- Since this study is based on the cohort study, please provide the detailed information about ethical approval.

Minor comments

- Please spell out MDRD in line 146.
- In Figure 2, please revise the legend to use “A”, “B”, and “C” to correspond with the graphs in the figure. Additionally, please ensure that the result section describing Figure 2 (around line 345 to 347) also uses “A”, “B”, and “C” to accurately describe the findings.
- The unit of eGFR in line 337 and 478 is “min/min/1.73m²”. Please provide correct unit (ml/min/1.73m²)
- Supplementary tables should be provided in excel file or csv file.

Response to Reviews: Dubin et al. Proteomics of CKD Progression in the Chronic Renal Insufficiency Cohort

Page numbers refer to tracked version

Reviewer 1:

1. The introduction and discussion emphasize the need for a prognostic equation that includes modifiable factors and lists this as a limitation of the KFRE equation. I fully agree that both - good prognostic equations and the identification of modifiable / treatable factors – are very important challenges that require our attention. However, it is not clear to me why these need to be addressed in the same equation. Some clarification on that aspect or a rephrasing might be helpful.

*Response: Thank you for this important question. We believe that the prognostic equation should consist of modifiable risk factors so it can be used to monitor whether any treatments initiated are effective. We agree with the reviewer that there are treatable risk factors that can be identified separately from the prognostic equation. We have added the following as clarification in the **Introduction, page 5**:*

Personalized prognostic equations for CKD progression that consist of modifiable biological factors could be used to monitor response to medical treatment. For example, a prognostic equation for cardiovascular risk that consisted of modifiable protein risk factors accurately predicted which patients were still at risk for poor outcomes and might benefit from more specialized therapies.¹

2. Furthermore, the druggability was followed up on for the elastic net selected proteins. Elastic nets are not designed to identify all putatively causal variables and will for example pick one of two highly correlated variables (to a certain degree). A more conclusive list for potential downstream studies would be the same protein selection used for the MR screen.

*Response: Thank you, we hope to provide clarification as follows. We researched pharmaceutical agents for each of the proteins in a) the list of proteins selected by significance and effect size of hazard ratio for the primary outcome in CRIC, that also validated in ARIC; and b) the proteins that were selected by elastic net (EN) in CRIC and included in the 65-protein risk model. All 76 proteins selected by either a) or b) were candidates for Mendelian randomization, and we researched the druggability for all 76, as well. Proteins, along with any relevant drugs, selected by method a) are listed in **Table 1** and **Supplemental Tables 5,6**. Thus, as we understand the reviewer's comment, we have already performed the proposed wider screen for druggable targets.*

3. A list of all analyzed proteins with some basic annotations (IDs, full names, etc) would be helpful supplementary material. Further the CVs from the 129 duplicates could be a nice resource to other SomaScan studies. This as well as the existing supplementary table 1 would benefit from a non-pdf format like xlsx.

*Response: Thank you, we have added **Supplemental Table 1**, which includes all aptamers, CV's and MAD values. The Supplemental Table 1 as well as all other Supplemental Tables are now presented in an Excel format, as suggested by Reviewers 1 and 2.*

4. Protein level transformation: The parallel presentation of log2 and MAD unit analyses was slightly confusing to me. E.g. the methods mention that HRs are given in log2 units (implying only in log2)

- Consider presenting log2 and MAD unit HR in merged tables to ease the comparison and condense the supplementary materials. The actual MADs would also be a great addition to the proposed added supplementary table that lists all proteins.

*Response: Thank you, we have added MAD unit HR to **Supplemental tables 7,8 and 11,12** so now the HRs per log2 and MAD are shown side by side in the same table, as the reviewer recommended. The new **Supplemental Table 1** lists the MAD values for all proteins, as the reviewer recommended.*

- More importantly, MAD scaling and a log2-transformation have different impacts on the distribution of proteins. What is the impact of the two transformations on skewness and similar parameters of the protein distributions? If robustness was the major concern driving the decision for MAD scaling, did you consider a rank-based inverse normal transformation?

Response: Thank you for making these important points. We respectfully wish to explain why we chose to use MAD and log2 for our study. In our analysis of individual protein associations with outcomes, our goal was to prioritize which proteins to replicate in ARIC and then carry these to Mendelian randomization analysis. We chose to rank protein associations by using HR per median absolute deviation (MAD) in order to put all proteins on the same scale, akin to conventional (mean, standard deviation) standardization, but using the more robust measures of centrality and spread (median, MAD). We have not performed per-protein tests of normality since, as noted, normally distributed predictors are not required for either our univariate or multivariate (elastic net) analytic approaches.

We agree, it is important to consider that there are different transformations such as rank-based inverse normal transformation. We did not use rank-based inverse normal transformation, which is intended to produce a normal distribution, for several reasons. Since it is rank-based, it may cause loss of information. We were not seeking to establish symmetric / normal distributions for each protein. Additionally, HR per log2 and HR per MAD are more commonly used for biomarker research.

The reason we included log2 in our presentation of the protein associations in table format is that this unit is more familiar to readers than MAD unit and provides a better measure of effect size to compare our biomarkers to those previously published in epidemiological studies.

5. Refitting of the EN model: “After the final selection of proteins, to reduce bias in estimated regression coefficients, we refit the selected features for the EN model ...” Wouldn’t the refitting after the variable selection introduce bias / overfitting instead of removing it? Isn’t the penalty in the elastic net approach designed to limit the potential overfitting due to the high number of variables?

Response: The reviewer brings up excellent points that we wish to address. Refitting elastic net selected proteins (without penalization -- so-called 'relaxation') is intended to remove potential

coefficient biases incurred by penalized fitting procedures as first published by Meinshausen.² The reviewer is correct in noting that this could re-introduce overfitting and might increase variance. To avoid overfitting, we trained the elastic net model in an 80% training set and tested it in a 20% testing set and conducted cross-validation of the EN algorithm in a series of 80/20 splits. Similarly, we derived refit coefficients in the 80% training set and present the results for this model in the 20% testing set. C-statistics for the EN model and refit model compared were virtually identical. Ultimately, we chose to use the refit model because this is the convention of similar studies.

We have revised this sentence, **Methods page 12**, to provide references for the practice of refitting the EN model:

After the final selection of proteins, to reduce bias in estimated regression coefficients,² we refit the selected features for the EN model in a Cox regression model for the CKD progression survival outcomes and a logistic regression model for the binary eGFR decline outcome, as previously published.³

6. Significance testing:

a. The methods for the primary screen in CRIC lists both a Bonferroni threshold as well as a BH adjustment. For external validation you focused on the highest and lowest 20 HRs. Furthermore, some proteins that “narrowly missed” significance were reported. Results and discussion switch between FDR and Bonferroni reporting. In combination with three different adjustment levels this is challenging to follow. A more aligned multiple testing and validation design would be desirable.

Response: Thank you for pointing this out. We did use different statistical significance thresholds for the primary screen in CRIC than for the validation of proteins in ARIC. We believed it was necessary to use $FDR < 0.05$ to allow for a broader array of proteins in the initial screen, and that it was appropriate to use the stricter significance level corrected for multiple testing as a criterion for replication. Please see explanation of this under (b), below.

b) Methods: “Bonferroni-corrected statistical significance levels were reported after adjusting for 4638 comparisons among proteins resulting in a nominal significance level of $p < 1.0 \times 10^{-5}$.” Technically it is 4830 comparisons, right? I agree with the choice to account for the number of proteins. Why is the threshold called nominal when it reflects a full Bonferroni correction? Did you apply the rounded threshold or $0.05/4638$?

*Thank you, we agree that clarification is needed in several sections that cite different thresholds for statistical significance. For the analyses of individual proteins in CRIC, we chose the $FDR < 0.05$ as a cutoff for screening in order to minimize type II error at the screening stage. We report p-values, because in fact many of the proteins we present do meet this criterion. The Bonferroni level of significance in CRIC was calculated as $0.05/5000$ (rather than $0.05/4830$, as the reviewer noted). The criterion for replication in ARIC was based on 40 tests, based on 40 proteins we were attempting to validate. We have clarified in the following section of the **Methods, page 10**:*

Evaluating each individual protein was a preliminary step, prior to determining which proteins to replicate externally, and then to examine with Mendelian randomization. In order to rank individual proteins by strength of association with the outcome, we

employed MAD standardization because it is more robust than log₂ standardization for skewed predictors. We chose to select 'top hits' from among the protein associations meeting a significance threshold of FDR <0.05, rather than Bonferroni significance, to minimize type II error at the screening stage. The Benjamini-Hochberg (BH) method was used to control the false discovery rate (FDR) at 5%.^{4, 5} We then selected protein 'top hits' by effect size per MAD unit. We present these top hits in tables using HR per log₂ to illustrate effect sizes on a scale more commonly used in epidemiology than MAD. Presentation tables also include the p-value in order to illustrate that most of these proteins meet Bonferroni-corrected statistical significance level ($p < 1.0 \times 10^{-5}$ after adjusting for ~5000 tests).

*In the **Methods, page 13**, we have clarified:*

The statistical criterion for validation was a Bonferroni p-value of $<(0.05/40)$ or <0.00125 , based on correcting for 40 proteins carried forward for validation.

*We have simplified the following **Results** section (**page 16-17**) describing individual protein associations. We limit our discussion of proteins associations to $FDR < 0.05$ in CRIC and refer to HR per log₂ for the purpose of framing effect sizes in terms comparable to typical biomarker studies:*

Associations of Individual Proteins with the Primary Outcome in CRIC and ARIC

Associations of individual proteins with the primary outcome ($\geq 50\%$ eGFR decline or ESRD within 10 years) are visualized in **Figure 2** as Volcano plots, shown unadjusted (**Figure 2A**), adjusted for eGFR (**Figure 2B**) and fully adjusted (**Figure 2C**). Among the 4638 proteins investigated, in fully adjusted analyses, 330 proteins (7.1% of all proteins measured) were associated with primary renal outcome at FDR significance ($q < 0.05$). We identified numerous proteins associated with higher risk of the primary outcome. Whereas only 1 of the previously reported 17 KRIS proteins had fully adjusted log₂ HR > 2 , 14 additional proteins with fully adjusted HRs between 2 to 5 were newly identified in this study (**Figure 2**). The top 20 proteins with the largest HR per log₂, listed with their biological functions and current drugs that target them, are shown in **Table 1**. We identified numerous proteins associated with lower risk ($HR < 1$), referred to in the literature as potentially protective, which are shown in **Figure 2 and Supplemental Table 7**. Protein associations are shown as HR per MAD unit in the Supplement (**Supplemental Tables 6, 7, 10-13**).

*For clarity, we deleted this sentence from **Results (page 17)**:*

Two additional proteins with $HR < 1$, epidermal growth factor receptor and afamin, narrowly missed the adjusted significance threshold in ARIC ($p < 0.00125$).

c. The primary analysis on individual proteins focused the reporting on the fully adjusted models. The pathway analysis starts out with the proteins associated in the eGFR-only-adjusted models. Is there a specific reason driving this switch?

Thank you, we offer the following explanation. Pathway analyses are typically performed on proteins using unadjusted associations with the outcome of interest. In this instance, we opted to adjust for eGFR to remove many proteins whose level primarily reflects low glomerular

filtration (so-called filtration markers), since lower renal function increases levels of small to medium sized solutes. Adjusting for comorbidities beyond eGFR prior to a pathway analysis would be unconventional and counterproductive in that it might hide the importance of a pathway such as fibrosis which is common to comorbidities such as diabetes and kidney disease.

We focused on full adjustment in the Results section on individual protein associations, because it is necessary to determine whether the protein association is independent of comorbidities in order to consider a protein as a potential biomarker that could have future clinical applications.

Minor:

- The abstract mentions the discrimination of the 65 protein model without including the actual statistic. Consider including a key discriminator.

*Thank you, we have added the C-statistic of the model to the **Abstract**:*

A 65-protein risk model for the primary outcome had excellent discrimination (C-statistic[95%CI] 0.862 [0.835, 0.889]), ~~similar to the Kidney Failure Risk Equation~~ and 14/65 proteins are druggable targets.

- Abstract: “We found 100 plasma proteins independently associated with the primary CKD progression outcome.” Here, the independently would imply to me that all 100 proteins are significant in a joint model. However, I believe this to be referring to the 100 individual protein analyses that were significant. Consider removing “independently”.

Thank you, we have removed ‘independently.’

- Consider switching nomenclature from ESRD to kidney failure according to KDIGO guidelines.

Thank you, we have replaced this nomenclature throughout.

- Methods – Participants: Add “prevalent” in the following? “We excluded 53 participants with ESRD.”

*Thank you, we have added ‘prevalent.’ (**Methods page 7**)*

- Methods – Study Outcomes: Why was the 2021 CKD EPI eGFR_{crea} equation instead of the 2021 CKD EPI eGFR_{crea-cys} equation used for sensitivity analyses? The second seems closer to the CRIC equation.

Response: Thank you, this is an important issue that we carefully considered and discussed with CRIC leadership. The 2021 CKD EPI eGFR_{cr-cys} equation is based on the internationally standardized cystatin assay that was adapted by Siemens in 2018. Cystatin values can be 20% higher in assays that use these newer reagents compared to the cystatin values obtained using older reagents.⁶ While the cystatin assays in CRIC were performed with older reagents, the CRIC estimated GFR equation was developed in a sizeable subgroup of CRIC participants who

had simultaneous measured GFR with iothalamate.⁷ We are thus confident that the CRIC eGFR equation is the most precise measure of eGFR and change in eGFR for CRIC participants. We chose not to use the CKD EPI 2021 cys-cr equation for sensitivity analyses, since the cystatin assay used in CKD EPI 2021 differed from the one used in CRIC. This decision to use the CKD EPI 2021 cr equation for the sensitivity analysis originated after careful deliberation within CRIC leadership, who are the intimately familiar with the eGFR and cystatin measures in the cohort.

- Methods: "To optimize the performance of these clinical models in CRIC, the coefficients of the variables of both clinical models were refit..." Was this done using the same test dataset as the elastic nets were trained on?

Response: Yes, we used the same testing set as for the elastic net models.

- How were the conditional cis-pQTL analyses conducted? And how were the CRIC and deCODE pQTL combined in the MR analysis?

*Response: We have added information about the conditional analyses (**Methods page 14**):*

Furthermore, we conducted the conditional association analysis within the candidate set with the GCTA-COJO software⁸ and selected the conditional significant variants with the p-value threshold 5×10^{-6} for the subsequent MR analysis.⁹

*Response: We did not combine the CRIC and deCODE pQTL. We considered them separately, and searched GWAS for pQTL found in either cohort. pQTLs found in deCODE are marked with asterisk in **Figure 3**.*

- For the eGFR MR a larger EA GWAS is available (N=1 million) from the CKDGen consortium (Stanzick et al. 2021; <https://www.nature.com/articles/s41467-021-24491-0>)

Response: The Reviewer makes an excellent point, and we appreciate it. We concur that assessment of MR in larger samples such as the study by Stanzick et al will be an important next step. We see that important pursuit as a follow up study (focused directly on MR analyses of proteomic correlates of kidney function), and one not directly within the limited scope of the current manuscript (where MR is not a primary purpose).

Acknowledging the importance of the reviewer's comment, we have added the following text to the Limitation and Future Directions section of our manuscript and now cite the important Stanzick paper:

Lastly, the present Mendelian randomization analyses may be augmented by utilizing a more comprehensive GWAS for renal function that includes a meta-analysis of CKD Genetics Consortium and UK Biobank.¹⁰

- Something went wrong in the formatting of references 34 and 35.

Response: Thank you, we have made this correction.

Reviewer #2:

In this manuscript, the author analyzed 3235 participants of CRIC study. They found 100 proteins which are independently associated with CKD progression outcome. These proteins were found to be associated with several biological pathways, including bone morphogenetic proteins, ephrin signaling, and prothrombin activation. The authors also developed multi-protein risk models for predicting CKD progression using machine learning and identified potential druggable targets. The study's methodology and experiments with a large number of cohort participants are carefully done and support the author's conclusions. However, the following concerns should be addressed before its publication at Nature communications.

Major comments

- The authors analyzed plasma samples from participants and developed a multi-protein risk model to predict CKD progression. However, it is unclear whether the changes in protein expression of plasma samples are truly linked to CKD progression that occurs in kidney tissue. Although the authors acknowledged this limitation in the Discussion section, they need to provide a more detailed plan to validate the proteomics information presented in their study.

*Response: Thank you for requesting this explanation of potential validation studies. We have added to the **Limitations, page 27**:*

The prognostic utility of the multi-protein risk score, and its capacity to reflect effects of medications, could be validated using samples from clinical trials involving kidney endpoints. The biological roles of specific proteins could be elucidated with animal models.

- In the method section, the authors explained very well regarding participants in this study, statistical analysis, and derivation of risk models. However, the method used for detecting protein expression in plasma using SomaScan Version 4.0 was not well explained. As proteome analysis is a critical part of this study, it is recommended to provide more details about SomaScan. Specifically, the authors could explain the sample preparation process, the equipment used in the study, and provide a simplified explanation of how SomaScan works to detect proteins. This additional information would help readers to better understand and replicate the proteomics analysis.

*Response: Thank you for this excellent suggestion. We have added to the **Methods, page 7**:*

SomaScan is an assay based on modified aptamers, which are chemically modified single strands of deoxyribonucleic acid ~40 nucleotides long, as binding reagents for target proteins.¹¹⁻¹⁶ Modified aptamers bind to proteins with high affinity similar to antibodies (lower limit of detection 10^{-15} moles per liter.)^{11, 13, 14} "Pull-down" studies, in which the aptamer-protein complexes were isolated and the identities of the bound proteins were verified by targeted mass spectrometry and gel electrophoresis, have been performed for 920 proteins among 1305 proteins in a previous version of the assay.¹⁵ These studies showed that > 95% of aptamers correctly targeted the intended proteins (for those proteins in concentrations sufficient to be detected by mass spectrometry). The samples on the SomaScan assay are run at three different dilutions to assay each analyte within its linear range of concentrations. The assay results are quantified on a hybridization microarray and reported in RFU. SomaLogic has

procedures for data calibration, standardization and internal controls, typical of microarray technologies.

- Regarding Figure 2, because protein name in volcano plot is quite small and its low resolution of the graph, it is tough to understand easily for the readers. In Supplementary Figure 1, protein name is also quite small.

Response: Thank you, we have enlarged the font as well as enhanced the resolution of these figures.

- The authors identified druggable targets among the proteins associated with CKD progression and provided information about the drug names in the Supplementary Tables, including Tables 12 and 13. In order to facilitate their clinical translation in the future, it would be useful to provide additional information about these drugs, such as whether they have already been approved for other diseases or are still in the development phase.

*Response: Thank you for this excellent suggestion, we have added this information to **Supplemental Tables 5,6,13,14**.*

- Related to the comment above, the authors state that there is correlation between plasma samples protein and CKD progression, and it might be a biomarker or druggable target. Is it applicable to the clinic? Please provide the potential strategy for detecting or preventing CKD progression.

*Response: Thank for bringing attention to clinical application of the protein risk model. We have added to **Discussion, page 26**:*

Clinicians might use the protein model not only to identify patients at higher risk of kidney failure, but also to monitor patients' response to lifestyle and medication changes. Showing the patient that his or her risk score has improved could improve compliance with medications.

- Since this study is based on the cohort study, please provide the detailed information about ethical approval.

*Response: We have added the following to **Methods, page 7**:*

The CRIC study was approved by the Institutional Review Boards of the participating centers and the research was conducted in accordance with the principles of the Declaration of Helsinki. All study participants provided written informed consent.

Minor comments

- Please spell out MDRD in line 146. *Thank you, we have made this correction.*
- In Figure 2, please revise the legend to use "A", "B", and "C" to correspond with the graphs in the figure. Additionally, please ensure that the result section describing Figure 2 (around line 345 to 347) also uses "A", "B", and "C" to accurately describe the findings. *We have made these corrections.*
- The unit of eGFR in line 337 and 478 is "min/min/1.73m²". Please provide correct unit

(ml/min/1.73m²) We have made this correction.

· Supplementary tables should be provided in excel file or csv file. We have now submitted the Excel versions.

References

1. Williams SA, Ostroff R, Hinterberg MA, Coresh J, Ballantyne CM, Matsushita K, Mueller CE, Walter J, Jonasson C, Holman RR, Shah SH, Sattar N, Taylor R, Lean ME, Kato S, Shimokawa H, Sakata Y, Nochioka K, Parikh CR, Coca SG, Omland T, Chadwick J, Astling D, Hagar Y, Kureshi N, Loupy K, Paterson C, Primus J, Simpson M, Trujillo NP and Ganz P. A proteomic surrogate for cardiovascular outcomes that is sensitive to multiple mechanisms of change in risk. *Sci Transl Med.* 2022;14:eabj9625.
2. Meinshausen N. Relaxed Lasso. *Computational Statistics and Data Analysis.* 2007;52:374-393.
3. Deo R, Dubin RF, Ren Y, Murthy AC, Wang J, Zheng H, Zheng Z, Feldman H, Shou H, Coresh J, Grams M, Surapaneni AL, Bhat Z, Cohen JB, Rahman M, He J, Saraf SL, Go AS, Kimmel PL, Vasan RS, Segal MR, Li H and Ganz P. Proteomic cardiovascular risk assessment in chronic kidney disease. *Eur Heart J.* 2023;44:2095-2110.
4. Harrell FE, Jr., Califf RM, Pryor DB, Lee KL and Rosati RA. Evaluating the yield of medical tests. *JAMA.* 1982;247:2543-6.
5. Hochberg Y and Benjamini Y. More powerful procedures for multiple significance testing. *Stat Med.* 1990;9:811-8.
6. Benoit SW, Kathman T, Patel J, Stegman M, Cobb C, Hoehn J, Devarajan P and Nehus EJ. GFR Estimation After Cystatin C Reference Material Change. *Kidney Int Rep.* 2021;6:429-436.
7. Anderson AH, Yang W, Hsu CY, Joffe MM, Leonard MB, Xie D, Chen J, Greene T, Jaar BG, Kao P, Kusek JW, Landis JR, Lash JP, Townsend RR, Weir MR, Feldman HI and Investigators CS. Estimating GFR among participants in the Chronic Renal Insufficiency Cohort (CRIC) Study. *Am J Kidney Dis.* 2012;60:250-61.
8. Yang J, Lee SH, Goddard ME and Visscher PM. GCTA: a tool for genome-wide complex trait analysis. *Am J Hum Genet.* 2011;88:76-82.
9. Yang J, Ferreira T, Morris AP, Medland SE, Genetic Investigation of ATC, Replication DIG, Meta-analysis C, Madden PA, Heath AC, Martin NG, Montgomery GW, Weedon MN, Loos RJ, Frayling TM, McCarthy MI, Hirschhorn JN, Goddard ME and Visscher PM. Conditional and joint multiple-SNP analysis of GWAS summary statistics identifies additional variants influencing complex traits. *Nat Genet.* 2012;44:369-75, S1-3.
10. Stanzick KJ, Li Y, Schlosser P, Gorski M, Wuttke M, Thomas LF, Rasheed H, Rowan BX, Graham SE, Vanderweff BR, Patil SB, Program VAMV, Robinson-Cohen C, Gaziano JM, O'Donnell CJ, Willer CJ, Hallan S, Asvold BO, Gessner A, Hung AM, Pattaro C, Kottgen A, Stark KJ, Heid IM and Winkler TW. Discovery and prioritization of variants and genes for kidney function in >1.2 million individuals. *Nat Commun.* 2021;12:4350.
11. Brody EN, Gold L, Lawn RM, Walker JJ and Zichi D. High-content affinity-based proteomics: unlocking protein biomarker discovery. *Expert Rev Mol Diagn.* 2010;10:1013-22.
12. Ganz P, Heidecker B, Hveem K, Jonasson C, Kato S, Segal MR, Sterling DG and Williams SA. Development and Validation of a Protein-Based Risk Score for Cardiovascular Outcomes Among Patients With Stable Coronary Heart Disease. *JAMA.* 2016;315:2532-41.
13. Gold L, Ayers D, Bertino J, Bock C, Bock A, Brody EN, Carter J, Dalby AB, Eaton BE, Fitzwater T, Flather D, Forbes A, Foreman T, Fowler C, Gawande B, Goss M, Gunn M, Gupta S, Halladay D, Heil J, Heilig J, Hicke B, Husar G, Janjic N, Jarvis T, Jennings S, Katilius E, Keeney TR, Kim N, Koch TH, Kraemer S, Kroiss L, Le N, Levine D, Lindsey W, Lollo B, Mayfield W,

Mehan M, Mehler R, Nelson SK, Nelson M, Nieuwlandt D, Nikrad M, Ochsner U, Ostroff RM, Otis M, Parker T, Pietrasiewicz S, Resnicow DI, Rohloff J, Sanders G, Sattin S, Schneider D, Singer B, Stanton M, Sterkel A, Stewart A, Stratford S, Vaught JD, Vrkljan M, Walker JJ, Watrobka M, Waugh S, Weiss A, Wilcox SK, Wolfson A, Wolk SK, Zhang C and Zichi D. Aptamer-based multiplexed proteomic technology for biomarker discovery. *PLoS One*. 2010;5:e15004.

14. Rohloff JC, Gelinis AD, Jarvis TC, Ochsner UA, Schneider DJ, Gold L and Janjic N. Nucleic Acid Ligands With Protein-like Side Chains: Modified Aptamers and Their Use as Diagnostic and Therapeutic Agents. *Mol Ther Nucleic Acids*. 2014;3:e201.

15. Sun BB, Maranville JC, Peters JE, Stacey D, Staley JR, Blackshaw J, Burgess S, Jiang T, Paige E, Surendran P, Oliver-Williams C, Kamat MA, Prins BP, Wilcox SK, Zimmerman ES, Chi A, Bansal N, Spain SL, Wood AM, Morrell NW, Bradley JR, Janjic N, Roberts DJ, Ouwehand WH, Todd JA, Soranzo N, Suhre K, Paul DS, Fox CS, Plenge RM, Danesh J, Runz H and Butterworth AS. Genomic atlas of the human plasma proteome. *Nature*. 2018;558:73-79.

16. Williams SA, Kivimaki M, Langenberg C, Hingorani AD, Casas JP, Bouchard C, Jonasson C, Sarzynski MA, Shipley MJ, Alexander L, Ash J, Bauer T, Chadwick J, Datta G, DeLisle RK, Hagar Y, Hinterberg M, Ostroff R, Weiss S, Ganz P and Wareham NJ. Plasma protein patterns as comprehensive indicators of health. *Nat Med*. 2019;25:1851-1857.

REVIEWERS' COMMENTS

Reviewer #1 (Remarks to the Author):

Thank you very much for the excellent rebuttal that addresses all my suggestions.

PS: In the reviewer pdf it looks like the reporting summary is missing the signature.

Reviewer #2 (Remarks to the Author):

The authors have addressed all my questions and concerns raised previously.

I have no further questions.

Thus, I recommend publication of this manuscript in Nature communications.